# Genetic Potential and Inheritance Patterns of Physiological, Agronomic and Quality Traits in Bread Wheat under Normal and Water Deficit Conditions

**DOI:** 10.3390/plants11070952

**Published:** 2022-03-31

**Authors:** Mohamed M. Kamara, Medhat Rehan, Amany M. Mohamed, Rania F. El Mantawy, Ahmed M. S. Kheir, Diaa Abd El-Moneim, Fatmah Ahmed Safhi, Salha M. ALshamrani, Emad M. Hafez, Said I. Behiry, Mohamed M. A. Ali, Elsayed Mansour

**Affiliations:** 1Department of Agronomy, Faculty of Agriculture, Kafrelsheikh University, Kafr El-Sheikh 33516, Egypt; mohamed.kamara@agr.kfs.edu.eg (M.M.K.); emadhafez2012@agr.kfs.edu.eg (E.M.H.); 2Department of Plant Production and Protection, College of Agriculture and Veterinary Medicine, Qassim University, Burydah 51452, Saudi Arabia; 3Department of Genetics, College of Agriculture, Kafrelsheikh University, Kafr El-Sheikh 33516, Egypt; 4Seed Technology Research Department, Field Crops Research Institute, Agricultural Research Center, Giza 12619, Egypt; amanymahmoud751@gmail.com; 5Crop Physiology Research Department, Field Crops Research Institute, Agricultural Research Center, Giza 12619, Egypt; raniafarouk711@gmail.com; 6Soils, Water and Environment Research Institute, Agricultural Research Center, Giza 12112, Egypt; ahmed.kheir@arc.sci.eg; 7International Center for Biosaline Agriculture, Directorate of Programs, Dubai 14660, United Arab Emirates; 8Department of Plant Production (Genetic Branch), Faculty of Environmental Agricultural Sciences, Arish University, El-Arish 45511, Egypt; dabdelmoniem@aru.edu.eg; 9Department of Biology, College of Science, Princess Nourah bint Abdulrahman University, P.O. Box 84428, Riyadh 11671, Saudi Arabia; faalsafhi@pnu.edu.sa; 10Department of Biology, College of Science, University of Jeddah, Jeddah 21959, Saudi Arabia; smalshmrane@uj.edu.sa; 11Agricultural Botany Department, Faculty of Agriculture (Saba Basha), Alexandria University, Alexandria 21531, Egypt; said.behiry@alexu.edu.eg; 12Department of Crop Science, Faculty of Agriculture, Zagazig University, Zagazig 44519, Egypt; abd_lhamed@yahoo.com (M.M.A.A.); sayed_mansour_84@yahoo.es (E.M.)

**Keywords:** *Triticum aestivum*, cluster analysis, drought, gene action, heterosis, principal component analysis

## Abstract

Water scarcity is a major environmental stress that adversatively impacts wheat growth, production, and quality. Furthermore, drought is predicted to be more frequent and severe as a result of climate change, particularly in arid regions. Hence, breeding for drought-tolerant and high-yielding wheat genotypes has become more decisive to sustain its production and ensure global food security with continuing population growth. The present study aimed at evaluating different parental bread wheat genotypes (exotic and local) and their hybrids under normal and drought stress conditions. Gene action controlling physiological, agronomic, and quality traits through half-diallel analysis was applied. The results showed that water-deficit stress substantially decreased chlorophyll content, photosynthetic efficiency (FV/Fm), relative water content, grain yield, and yield attributes. On the other hand, proline content, antioxidant enzyme activities (CAT, POD, and SOD), grain protein content, wet gluten content, and dry gluten content were significantly increased compared to well-watered conditions. The 36 evaluated genotypes were classified based on drought tolerance indices into 5 groups varying from highly drought-tolerant (group A) to highly drought-sensitive genotypes (group E). The parental genotypes P_3_ and P_8_ were identified as good combiners to increase chlorophyll b, total chlorophyll content, relative water content, grain yield, and yield components under water deficit conditions. Additionally, the cross combinations P_2_ × P_4_, P_3_ × P_5_, P_3_ × P_8_, and P_6_ × P_7_ were the most promising combinations to increase yield traits and multiple physiological parameters under water deficit conditions. Furthermore, P_1_, P_2_, and P_5_ were recognized as promising parents to improve grain protein content and wet and dry gluten contents under drought stress. In addition, the crosses P_1_ × P_4_, P_2_ × P_3_, P_2_ × P_5_, P_2_ × P_6_, P_4_ × P_7_, P_5_ × P_7_, P_5_ × P_8_, P_6_ × P_8_, and P_7_ × P_8_ were the best combinations to improve grain protein content under water-stressed and non-stressed conditions. Certain physiological traits displayed highly positive associations with grain yield and its contributing traits under drought stress such as chlorophyll a, chlorophyll b, total chlorophyll content, photosynthetic efficiency (Fv/Fm), proline content, and relative water content, which suggest their importance for indirect selection under water deficit conditions. Otherwise, grain protein content was negatively correlated with grain yield, indicating that selection for higher grain yield could reduce grain protein content under drought stress conditions.

## 1. Introduction

Wheat (*Triticum aestivum* L.) is one of the most important cereal crops [1,2,3]. Wheat grain is a vital source that supports humans with calories, carbohydrates, protein, and vitamins [4,5]. Moreover, its straw is utilized in animal feeding and other industrial products [6,7]. The global population is expected to continue growing, and wheat demand is expected to rise, particularly in the face of global crises such as pandemics and wars. Consequently, considerable increase in wheat production is required tremendously to ensure food security [8]. Notwithstanding, its production is constrained by recent climate change, particularly in arid regions [9,10]. For instance, extreme climatic events, such as temperature rising and precipitation fluctuations, are expected to become more severe and frequent [11,12,13]. Drought stress is a harsh environmental factor that devastatingly affects global wheat production [14,15]. Nearly more than 50% of the cultivated area of wheat worldwide is subjected to frequent drought stress [16]. Moreover, global urbanization and industrialization have increased the pressure on freshwater resources [17]. Thus, water scarcity and drought problems are expected to worsen, which will negatively impact wheat production [18,19]. 

Water deficit reduces nutrients uptake, leaf water content, and photosynthesis, which deleteriously reflect on plant growth and productivity [20,21,22]. Furthermore, drought stress induces oxidative stress by raising the production of reactive oxygen species (ROS) [23,24,25]. The ROS damages nucleic acids, photosynthetic pigments, and membrane lipids as well as restricts the metabolism. As a result, there is a substantial reduction in number of leaves, leaf area, plant height, and grain weight, resulting in lower grain yield [26,27,28]. Enzymatic activities as catalase (CAT), peroxides (POD), and superoxide dismutase (SOD) are elevated under water deficit conditions to scavenge ROS and preserve cells from oxidative stress [29,30]. Additionally, increasing proline accumulation has been depicted in the plants that are exposed to drought stress [31,32,33]. It is an important osmoregulator that has an effective role in membrane stabilization to mitigate the injurious effect of water shortage [34,35]. Information on the inter-trait associations among grain yield and other physiological traits enhances the efficacy of breeding programs employing proper traits as selection criteria under drought stress and non-stressed conditions. Traits such as chlorophyll content, relative water content, proline content, and antioxidant activities could be utilized as secondary traits for screening drought-tolerant genotypes in breeding programs [24,36]. 

Grain quality of bread wheat is essential in terms of nutritional benefit and economics, accordingly, it receives increasing attention [37]. It is affected by genotypes, environments, and their interaction [38,39,40]. Drought is a crucial environmental factor that impacts the quality traits of wheat. Water deficit decreases photosynthesis, promotes leaf senescence, and limits the amount of assimilation, which causes a reduction in carbohydrate contents and total protein [41,42]. It also affects the nitrogen and carbohydrate assimilation rates, which can lead to significant changes in chemical composition, protein content, and starch granule size [42,43]. A negative association was reported between grain yield and grain protein content in wheat [44]. Hence, boosting grain yield and protein content is decisive for developing a high-quality wheat industry and ensuring nutrition and food security [45,46].

Combining the ability and nature of gene action governing agronomic, physiological, and quality traits in wheat could assist in determining merit parents for crossing and promising recombinants in breeding for drought tolerance [47,48,49]. Diallel mating design is a valuable biometric tool to study the general (GCA) and specific (SCA) combining ability effects and gene action in studied traits [50]. It also helps in selecting hybrids or parents for effective breeding either under normal or stressed conditions [51].

The aims of this study were (i) to assess the performance of eight wheat genotypes and their 28 F_1_ crosses for physiological, agronomic, and quality traits and exploring their diversity in drought tolerance; (ii) to explore the combining ability and type of gene action regulating the inheritance of the evaluated traits under water deficit and well-watered conditions; and (iii) to study the interrelationships among tested traits under drought conditions.

## 2. Materials and Methods

### 2.1. Hybridization and Experimental Site

Eight diverse wheat genotypes were selected based on origin diversity and drought tolerance from an earlier screening trial during the growing season of 2018 to 2019 (unpublished data). The used parents comprised four local cultivars, and four exotic genotypes (three from CIMMYT and one genotype from ICARDA). The pedigree of selected parents is shown in Appendix A. Half-diallel mating design (8 × 8) was made to generate 28 F_1_ hybrids during the growing season of 2019 to 2020. The parental genotypes and their cross combinations were assessed in two adjacent irrigation experiments at the Experimental Farm, Faculty of Agriculture, Kafrelsheikh University (31°6′ N, 30°56′ E), Egypt, during the growing season of 2020 to 2021. The 2 experiments were separated by a 5-m wide alley to prevent water leakage. The first experiment (normal condition) was irrigated 5 times throughout the season using the region’s standard practice, totaling approximately 450 mm, while the second experiment was irrigated twice throughout the whole season, with a total of approximately 190 mm, providing water-deficit conditions. The experimental site has an arid climate with an average annual rainfall of ∼80 mm. The meteorological data (i.e., maximum and minimum temperatures, solar radiation, and precipitation) over the growing season are presented in Figure 1. Soil properties of the experimental site are presented in Appendix A. The analysis revealed that the soil is clay throughout the profile (15.3% sand, 33.2% silt, and 51.5% clay). Randomized Complete Block Design (RCBD) with three replications was applied for each experiment. Each genotype was sown in 2 rows 2 m-long, with 0.30 m spacing between rows and 0.15 m between the plants. Phosphorus, potassium, and nitrogen fertilizers were performed at rates of 35 kg P_2_O_5_ ha^−1^, 57 kg K_2_O, and 180 kg N ha^−1^, respectively. The other agricultural practices comprising sowing date and weed, pest, and disease control were applied following the standard for wheat commercial production.

### 2.2. Measured Traits

#### 2.2.1. Physiological and Biochemical Traits

##### Chlorophyll Content and Chlorophyll Fluorescence

About 1 g fresh weight of mixed leaves was homogenized in 5 mL of 85% cold acetone and centrifuged. The extract was diluted to the appropriate volume before the optical density was determined at 663 and 647 nm [52]. The following equations were applied to calculate the chlorophyll content of the samples as mg/g fresh weight: Chlorophyll *a* = 11.79 E_663_ − 2.29 E_647_, Chlorophyll *b* = 20.05 E_647_ − 4.77 E_663_

The parameters of chlorophyll fluorescence were assessed in upper fully expanded leaf tissue using a portable Optic-Science OS-30p+ fluorometer (Opti-Sciences, Inc., Hudson, NH, USA). Calculations of maximum PS II Fv/Fm quantum yield were performed using the formula of Maxwell and Johnson [53]: Fv/Fm = (Fm − F0)/Fm.

##### Relative Water Content (RWC)

Leaf RWC was determined as outlined by Barrs and Weatherley [54]. Fresh weight (FW) of leaves was determined, then they were immersed in water for 5 h, and the turgid weights (TW) were recorded. Then in an oven at 80 °C for 24 h, the samples were dried and dry weight (DW) was determined. The RWC was calculated as follows: RWC = ((FW − DW)/(TW − DW)) × 100

##### Determination of Proline Content and Antioxidant Enzyme Activities

Proline content was determined as outlined by Bates, et al. [55]. Utilizing mortar and pestle, leaf samples (0.5 g) were homogenized in 5 mL of sulphosalicylic acid (3%). Almost 2 mL of extract was placed in a tube, and then 2 mL of ninhydrin reagent and 2 mL of glacial acetic acid were included. In a water bath at 100 °C for 60 min, the reaction mixture was boiled. After cooling the reaction mixture, 6 mL of toluene was included and then transferred to a separating funnel. After careful mixing, the chromophore including toluene was separated, and absorbance was read at 520 nm in a spectrophotometer against toluene blank. Proline concentration was recorded utilizing a calibration curve and expressed as mg proline g/FW.

The activities of catalase (CAT), peroxidase (POD), and superoxide dismutase (SOD) were determined as outlined by Aebi, et al. [56], Vetter, et al. [57], and Beauchamp and Fridovich [58], respectively. Fresh leaf samples (0.5 g) were homogenized in 5 mL of 50 mM cold K-phosphate buffer (pH 7.8). The homogenates were centrifuged for 20 min at 10,000× *g* at 4 °C. The supernatant was utilized to measure the antioxidant enzyme activity as (Units mg^−1^ protein).

#### 2.2.2. Agronomic Traits

Plant height (cm) was recorded as the distance from the soil surface to the tip of the spike, excluding awns. Number of grains/spike and spike length (cm) were recorded from 10 main spikes chosen randomly from each plot. Thousand grains weight (g) was evaluated as the weight of 1000 grains. Ten guarded pants from each plot were harvested, dried, and threshed, and the grain yield per plant (g) was determined.

#### 2.2.3. Grain Quality Traits

The grain quality traits were measured on samples taken from the grain bulk of each genotype. The grains samples (~20 g) were grounded to fine powder to pass through 2 mm mesh. Finally, the powder was used in the analysis of total crude protein and carbohydrate content. Total nitrogen content was analyzed utilizing the micro-Kjeldahl method, and then total crude protein % was computed by multiplying total N% by 5.85 [59]. Total carbohydrate in grinded wheat grains was analyzed as described in the method of Dubois, et al. [60]. Wet and dry gluten percentages were determined by hand-washing weighted meal samples according to the standard method of Pleshkov [61] until the starch was not detected in the washing water, then dried and weighed in grams.

### 2.3. Drought Tolerance Indices

Tolerance indices were calculated to identify potentially drought-tolerant genotypes. Geometric mean productivity (GMP) 
$=\sqrt{\mathrm{Ys}\;\times \;\mathrm{Yp}}$
 [62], mean productivity (MP) = (Y_s_ + Y_p_)/2 [63], yield index (YI) = Y_s_/Ȳ_s_ [64], and stress tolerance index (STI) = (Y_s_ × Y_p_)/(Ȳ_p_)^2^ [62]. Cluster analysis was performed based on tolerance indices to discriminate the tested genotypes according to their drought tolerance [65]. The cluster and principal component analyses were applied utilizing R statistical software version 4.4.1, library *factoextra* [66].

### 2.4. Statistical Analysis

Combining ability analysis was performed following Griffing’s method-2 model-1 [67]. Principal component analysis (PCA) and heatmap were applied using averages of the physiological, agronomic, and grain quality traits to explore the relationships among them using R statistical software.

## 3. Results

### 3.1. Diallel Analysis

Significant differences among genotypes (G), parents (P), F_1_ crosses (C), and P vs. C were detected for most evaluated traits under both conditions (Table 1). The mean squares of GCA and SCA were highly significant for all studied traits under both well-watered and stressed conditions. The ratio of GCA/SCA was more than the unity for all evaluated traits, except chlorophyll *a*, proline content, and catalase activity (CAT) under normal conditions; total chlorophyll content; the photosynthetic efficiency (Fv/Fm); peroxidase (POD) and superoxide dismutase (SOD) activities under water deficit conditions; and carbohydrates content under both conditions.

### 3.2. Mean Performance of the Evaluated Parents and Their Cross Combinations

#### 3.2.1. Physiological and Biochemical Traits

Drought stress significantly reduced chlorophyll *a* by 31.4% (Figure 2A and Figure 3A). The highest mean values were assigned for P_6_, P_8_, P_1_ × P_3_, P_1_ × P_4_, P_1_ × P_5_, P_1_ × P_6_, P_1_ × P_8_, P_2_ × P_6_, P_2_ × P_7_, P_2_ × P_8_, P_3_ × P_6_, P_3_ × P_8_, and P_4_ × P_8_ under water deficit conditions (Appendix A). Similarly, a water deficit declined chlorophyll *b* by 22.9% (Figure 2B and Figure 3A). The parents P_3_ and P_8_ and the hybrids P_2_ × P_4_, P_2_ × P_5_, P_2_ × P_8_, P_3_ × P_5_, P_4_ × P_5_, P_3_ × P_8_, P_4_ × P_6_, P_6_ × P_7_, and P_7_×P_8_ proved the highest values under water deficit conditions (Appendix A). The total chlorophyll content declined by 28.6% due to water limitation (Figure 2C and Figure 3A). The highest values were assigned for P_8_, P_3_ × P_4_, P_3_ × P_5_, P_3_ × P_8_, and P_6_ × P_7_ under stressed conditions (Appendix A). Likewise, photosynthetic efficiency (Fv/Fm) significantly decreased by 24.1% under water deficit (Figure 2D and Figure 3A). The genotypes P_1_, P_2_, P_3_, P_7_, P_1_ × P_6_, P_3_ × P_5_, P_3_ × P_8_, and P_4_ × P_8_ displayed the highest values under stressed conditions (Appendix A). Relative water content (RWC) was also depressingly impacted by drought stress; it decreased by 16.3% under water deficit (Figure 2E and Figure 3A). The highest values were given by, P_3_, P_8_, P_2_ × P_8_, P_3_ × P_8_, and P_4_ × P_5_ under stress conditions. Otherwise, water scarcity caused a considerable increase in proline content and the activities of antioxidant enzymes: CAT, POD, and SOD by 90.2, 107.5, 155.7, and 47.4%, respectively, compared to well-watered conditions (Figure 2F–I and Figure 3A). The genotypes P_3_, P_8_, P_3_ × P_5_, P_3_ × P_6_, and P_3_×P_8_ displayed the maximum values of proline content under stressed conditions (Appendix A). The highest values of CAT were recorded by P_5_, P_7_, P_2_ × P_3_, P_2_ × P_8_, P_5_ × P_7,_ and P_6_ × P_7_ under water deficit conditions (Appendix A). Regarding POD, the genotypes P_2_, P_6_, P_2_ × P_3_, P_2_ × P_6_ P_5_ × P_7,_ and P_6_ × P_8_ exhibited the largest values under stressed conditions (Appendix A). The genotypes P_1_, P_6_, P_1_ × P_8_, P_3_ × P_6_, and P_6_ × P_8_ recorded the highest SOD under water deficiency conditions (Appendix A).

#### 3.2.2. Agronomic Traits

Plant height was significantly affected by water deficit; it decreased by 12.3% compared to well-watered conditions (Figure 3B and Figure 4A). The genotypes P_1_, P_3_, P_4_, P_8_, P_1_ × P_8_, P_3_ × P_5_, and P_6_ × P_7_ possessed the tallest plants, whereas the shortest ones were P_2,_ P_5_, and P_2_ × P_6_ under water scarcity conditions (Appendix A). Likewise, the spike length was significantly reduced by 19.0% due to the decrease in the amount of irrigation water applied (Figure 3B and Figure 4B). The genotypes P_2_, P_4_, P_8_, P_1_ × P_8_, P_2_ × P_8_, P_3_ × P_8_, P_4_ × P_7_, and P_4_ × P_8_ had the longest spike under stress conditions (Appendix A). The number of grains/spike also declined by 20.0% due to water shortage conditions (Figure 3B and Figure 4C). The genotypes P_2_, P_4_, P_8_, P_2_ × P_4_, and P_3_ × P_8_ had the highest number of grains/spike under water scarcity conditions (Appendix A). Likewise, the water drought treatment dropped 1000 grain weight by 19.9% (Figure 3B and Figure 4D). The heaviest 1000-grain weight was assigned for P_1_, P_8_, P_1_ × P_3_, P_2_ × P_5_, P_3_ × P_7,_ and P_3_ × P_8_ under water scarcity conditions (Appendix A). Eventually, grain yield/plant was significantly reduced by 25.3% under water stress conditions (Figure 3B and Figure 4E). The genotypes P_3_, P_8,_ P_1_ × P_8_, P_2_ × P_4_, P_3_ × P_5_, P_3_ × P_8_, P_4_ × P_8_, and P_5_ × P_8_ displayed the highest grain yield under water deficiency conditions (Appendix A). 

#### 3.2.3. Grain Quality Traits

Water deficit significantly reduced carbohydrate content by 5.5% (Figure 3B and Figure 4F). The genotypes P_7_, P_8_, P_1_ × P_2_, P_2_ × P_8_, P_3_ × P_7_, P_4_ × P_8_ and P_6_ × P_7_ recorded the highest values under water scarcity conditions (Appendix A). Conversely, water deficit treatment significantly increased grain protein content by 22.0% (Figure 3B and Figure 4G). The genotypes P_1_, P_2_, P_5_, P_1_ × P_4_, P_2_ × P_5_, P_2_ × P_6_, P_5_ × P_7_, and P_5_ × P_8_ had the highest grain protein content under water deficit conditions (Appendix A). Likewise, wet gluten content was significantly affected by water deficit and also increased by 9.7% under deficit irrigation (Figure 3B and Figure 4H). The genotypes P_1_, P_2_, P_7_, P_1_ × P_7_, P_1_ × P_8_, P_2_ × P_3_, and P_2_ × P_8_ had the highest wet gluten content under water deficit conditions (Appendix A). Similarly, dry gluten content increased by 16.4% under water scarcity conditions (Figure 3B and Figure 4I). The highest values were obtained by the genotypes P_2_, P_5_, P_1_ × P_5_, P_3_ × P_7_, P_5_ × P_6_, and P_5_ × P_7_ under water deficit conditions (Appendix A).

### 3.3. Genotypic Classification Based on Drought Tolerance Indices

The hierarchical cluster classified the 36 evaluated genotypes into 5 groups according to their drought tolerance (Figure 5). Group (A) included two genotypes (P_3_ × P_8_ and P_4_ × P_8_) that possessed the highest tolerance indices (Appendix A); accordingly, they are considered highly drought-tolerant genotypes. The group (B) comprised of four genotypes (P_3_, P_8_, P_3_ × P_5_, P_6_ × P_7_, and P_5_ × P_6_) had high values; accordingly, they are deemed drought-tolerant genotypes. Similarly, group (C) consist of 14 genotypes with intermediate values of tolerance indices; hence, they are categorized as moderate drought-tolerant genotypes. Otherwise, eight genotypes in group (D) and seven genotypes in group (E) recorded the lowest values. Consequently, they are considered drought-sensitive and highly drought-sensitive genotypes, in the same order.

### 3.4. General Combining Ability (GCA) Effects

#### 3.4.1. Physiological and Biochemical Traits

The positive GCA estimates are pivotal for all traits, except negative plant height values are favorable. The GCA effects for measured traits differed significantly among the evaluated parents (Table 2). The desirable GCA effects of chlorophyll *a* were assigned for P_2_, P_3_, and P_4_ under normal conditions and P_1_, P_2_, P_6_, and P_8_ under water scarcity conditions. The favorable GCA effects for chlorophyll *b* and total chlorophyll were obtained by P_2_ under water deficit conditions, P_7_ under normal conditions, and P_3_ and P_8_ under both conditions. The highest GCA effects for Fv/Fm were expressed by P_2_ and P_7_ under non-stressed conditions and P_1_ and P_3_ under stress conditions. Moreover, significant GCA effects for RWC were detected for P_3_, P_6_, and P_8_ under both conditions. Additionally, P_3_ recorded the maximum GCA effects for proline content. The best combiners for CAT activity were P_6_ under normal conditions and P_2_, P_7_, and P_8_. Meanwhile, the highest GCA values for POD activity were exhibited by P_3_ and P_8_ under normal conditions, P_2_ under water deficit conditions, and P_6_ under both conditions. Furthermore, the highest GCA estimates for SOD activity were recorded by P_8_, followed by P_1_, under normal conditions, while the highest under stress conditions were for P_1_, followed by P_3_. 

#### 3.4.2. Agronomic Traits

The parents P_2_, P_5_, and P_6_ showed negative and significant effects towards dwarfness, while P_3_, P_4_, and P_8_ expressed positive and significant effects under both normal and water-stressed conditions. The highest GCA effects of spike length were recorded by P_4_ and P_8_ under both normal and stressful conditions. Positive GCA effects for the number of grains per spike were expressed by P_3_ under well-watered conditions, P_6_ and P_8_ under stress conditions, and P_2_ and P_4_ under both conditions. The best combiners for 1000-grain weight were demonstrated by P_4_ under well-watered conditions, P_3_ under drought stress conditions, and P_8_ under both conditions. The highest GCA for grain yield was obtained by P_1_ under well-watered conditions; P_6_ under drought stress conditions; and P_3_, P_4_, and P_8_ under both conditions. 

#### 3.4.3. Grain Quality Traits

The parents P_3_ and P_4_ had the highest GCA effects for carbohydrate content. Under both conditions, P_1_, P_2,_ and P_5_ exhibited the greatest positive and significant effects for grain protein content. Similarly, P_1_, P_2_, P_7,_ and P_8_ recorded the highest positive and significant effects for wet gluten content under both conditions. Finally, the highest GCA for dry gluten content was recorded by P_1_ and P_5_ under both conditions.

### 3.5. Specific Combining Ability (SCA) Estimates

#### 3.5.1. Physiological and Biochemical Traits

The SCA values for the cross combinations are presented in Table 3. Out of the 28 crosses, 5 under normal and 18 under stressed conditions recorded significantly positive SCA effects for chlorophyll *a.* The highest significant and positive SCA effects for chlorophyll *b* and total chlorophyll were obtained by P_2_ × P_4_, P_2_ × P_5_, P_2_ × P_8_, P_3_ × P_5_, P_3_ × P_8_, and P_6_ × P_7_ under both conditions. Respecting Fv/Fm, the SCA effects were positive and significant for the hybrids P_1_ × P_5_ and P_2_ × P_4_ under normal conditions, P_1_ × P_6_ and P_5_ × P_6_ under water scarcity conditions, and P_3_ × P_5_ and P_3_ × P_8_ under both conditions. The maximum and positively significant SCA values for RWC were exhibited by P_1_ × P_4_, P_1_ × P_6_, P_2_ × P_4_, P_2_ × P_7_, P_2_ × P_8_, P_3_ × P_5_, P_3_ × P_8,_ P_4_ × P_5,_ P_4_ × P_8_, and P_6_ × P_7_ under both normal and water-stressed conditions. For proline content, high SCA effects were obtained by P_1_ × P_6_, P_3_ × P_7_, P_4_ × P_6_, P_4_ × P_8_, P_5_ × P_6_, and P_5_ × P_8_ under stress conditions and P_3_ × P_5_ and P_3_ × P_8_ under both conditions. In the case of the activities of antioxidant enzymes, the cross combinations P_1_ × P_6_, P_2_ × P_3_, and P_3_ × P_8_ were the best specific combiners for CAT, P_1_ × P_2_, P_2_ × P_3_, P_3_ × P_8_, and P_5_ × P_7_ for POD and P_1_ × P_8_, P_2_ × P_3_, P_2_ × P_4_, P_2_ × P_7_, P_3_ × P_6_, and P_6_ × P_8_ for SOD under both conditions. 

#### 3.5.2. Agronomic Traits

High significant and negative SCA effects for plant height were recorded by P_1_ × P_3_, P_1_ × P_4_, P_1_ × P_6_, P_1_ × P_7_, P_2_ × P_3_, P_2_ × P_6_, P_2_ × P_7_, P_2_ × P_8_, P_3_ × P_4_, P_3_ × P_6_, P_4_ × P_5,_ and P_5_ × P_7_. In contrast, the highest positive SCA effects were recorded by P_1_ × P_2_, P_1_ × P_8_, P_2_ × P_4_, and P_6_ × P_7_ under both conditions. High significant and positive SCA estimates for spike length were revealed by P_3_ × P_5_, P_3_ × P_8_, P_5_ × P_6_ and P_6_ × P_7_ under both conditions. The crosses P_1_ × P_8_, P_2_ × P_4_, P_2_ × P_7_, P_3_ × P_5_, P_3_ × P_8_, P_4_ × P_5_, P_5_ × P_6_ and P_6_ × P_7_ expressed the largest positive and significant SCA values for number of grains per spike. Regarding 1000-grain weight, the hybrids P_2_ × P_5_, P_3_ × P_7_, P_3_ × P_8_ and P_6_ × P_7_ under both conditions displayed the highest positive SCA values. Additionally, the highest significant and positive SCA effects for grain yield per plant were obtained by P_1_ × P_2_, P_1_ × P_5_, P_1_ × P_8_ and P_3_ × P_7_ under normal conditions, P_4_ × P_8_ and P_5_ × P_6_ under water deficit conditions and P_2_ × P_4_, P_3_ × P_5_, P_3_ × P_8_ and P_6_ × P_7_ under both conditions. 

#### 3.5.3. Grain Quality Traits

The hybrids P_1_ × P_2_, P_1_ × P_5_, P_2_ × P_8_, P_3_ × P_7_, P_3_ × P_8_, P_4_ × P_5_, and P_4_ × P_6_ possessed significantly positive SCA effects for carbohydrate content. Similarly, the hybrids P_1_ × P_4_, P_2_ × P_3_, P_2_ × P_5_, P_2_ × P_6_, P_4_ × P_7_, P_5_ × P_7_, P_5_ × P_8_, P_6_ × P_8_, and P_7_ × P_8_ were identified to be good specific combiners for grain protein content. Likewise, the crosses P_1_ × P_6_, P_1_ × P_7_, P_1_ × P_8_, P_2_ × P_3_, P_2_ × P_4_, P_2_ × P_6_, P_2_ × P_8_, P_3_ × P_4_, P_3_ × P_7_, P_4_ × P_6_, P_5_ × P_6_, P_5_ × P_7_, P_6_ × P_7_, and P_7_ × P_8_ for wet gluten content and P_1_ × P_3_, P_1_ × P_5_, P_1_ × P_6_, P_2_ × P_3_, P_2_ × P_4_, P_3_ × P_7_, P_4_ × P_6_, P_5_ × P_6_, P_5_ × P_7_, and P_6_ × P_8_ for dry gluten content displayed the highest positive and significant SCA effects under both normal and water-stress conditions.

### 3.6. Interrelationship among Physiological, Agronomic, and Quality Traits

The first two principal components (PCAs) explained most of the variability, 60.80% (50.94% and 9.86% by PCA1 and PCA2). Hence, the two PCAs were utilized to construct the PC-biplot (Figure 6). Strong correlation was distinguished between grain yield and each of the following: chlorophyll *a*, chlorophyll *b*, total chlorophyll, Fv/Fm, RWC, proline content, plant height, spike length, 1000 grain weight, and number of grains per spike. In contrast, a negative association was detected between yield traits and each of the following: antioxidant enzyme activities (CAT, POD, and SOD), grain protein content, wet gluten content, and dry gluten content. Analogous results were deduced by the association heatmap as presented in Appendix A.

## 4. Discussion

Breeding drought-tolerant and high-yielding wheat genotypes has become increasingly important in order to sustain production in the face of continued population growth and climate change threats [68,69,70]. In the present study, highly significant variations were observed among the parental genotypes and their cross combinations for all studied physiological, agronomic, and quality traits under water deficit conditions. These findings revealed the existence of wide genetic variability in the evaluated materials, which could be exploited for developing drought-tolerant genotypes. This is in consonance with previous reports that demonstrated genetic variability in wheat genotypes for physiological [20,71,72,73], agronomic [51,74,75,76], and quality traits [77,78,79] under water-deficit stress conditions.

Drought is one of the most significant abiotic stresses limiting wheat production, particularly in arid environments [80]. The results elucidated that water scarcity caused substantial reductions in all assessed traits compared to well-watered treatment, except proline content, antioxidant enzyme activities (CAT, POD, and SOD), grain protein content, wet gluten content, and dry gluten content which significantly increased. Chlorophyll *a*, chlorophyll *b*, total chlorophyll content, photosynthetic efficiency (Fv/Fm), and relative water content of all tested genotypes were considerably reduced due to water deficit compared to the well-irrigated treatment. In this context, Arjenaki, et al. [81] and [82] deduced that drought stress declines water uptake from root system to the leaves [83]. Accordingly, it decreases the water-holding capacity and stomatal movement, which constrains chlorophyll synthesis, CO_2_ influx to leaves, and photosynthesis [76,84]. Furthermore, water deficit leads to the production of reactive oxygen species (ROS), including O_2_^−^, OH^−^, H_2_O_2_, and O_2_, in the plants, which causes oxidative damage and impairs cell functions. In addition, the accumulation of ROS causes chlorophyll degradation, the destruction of chloroplasts, and a reduction in photosystem II activity [85,86]. On the other hand, proline accumulation and the induction of CAT, POD, and SOD activities considerably increased in stressed wheat plants in comparison to non-stressed plants. These parameters are important defense mechanisms under drought stress [87,88]. Proline is an important osmolyte that protects plant cells against oxidative stress by osmotic adjustment, regulating cell redox balance, and protein stabilization [31,33]. SOD is the first defense wall in oxidative damage in the cells and plays a key role in the alteration of O_2_^−^ radicals to H_2_O_2_ and O_2_ [89]. POD converts H_2_O_2_ into H_2_O and oxygen, accordingly, assists the wheat plants in the detoxification mechanism against ROS species [14]. CAT participates in the conversion of H_2_O_2_ and plays a pivotal role in metabolism and signal recognition. 

Yield-contributing traits are the final products of physiological processes that occur at various development stages. The remarkable reductions in yield traits under water scarcity were caused by lack of absorbed water and inhibition of cell elongation and division [90]. Furthermore, the reduction in the number of grains/spike could be a result of sterility [91] or the abortion of immature embryos [92]. The decline in grain yield and its attributes in the present study was also elucidated by Morsy, et al. [75], Grzesiak, et al. [93], Mujtaba, et al. [94], Shamsi, et al. [95], Qayyum, et al. [23], and Sallam, et al. [15]

Grain quality is essential in terms of the nutritional benefit and economics of wheat. It varies according to genotypes, environments, and their interaction. Ozturk, et al. [77] elucidated that the environmental influences display great impacts on the variations of quality parameters of wheat grains than the genetic factors. The results indicated that water deficit conditions decreased carbohydrates content but increased grain protein content, wet gluten content, and dry gluten content compared to well-irrigated conditions. Generally, there is a negative association between grain yield and grain protein content, the reduction in irrigation reduces grain yield but increases the protein content. The improved protein content could be due to high accumulation rates of grain nitrogen and reduced rates of carbohydrates accumulation. Moreover, water deficiency diminishes carbohydrate synthesis and storage in the grain, enabling more protein accumulation per unit of starch [41,42,43]. In this respect, Elbasyoni, et al. [78], Saint Pierre, et al. [96], Ozturk, et al. [77], and Liu, et al. [97] reported enhanced grain protein and gluten content under water deficit compared to well-watered conditions in wheat.

A successful breeding program depends on the selection of suitable parental genotypes [98,99,100]. The GCA effects display valuable importance in identifying potential parents that could be employed in developing improved genotypes [101,102,103]. The results indicated that the parents P_3_ and P_7_ were recognized as promising parents for improving proline content under water deficit conditions. Moreover, P_2_, P_3_, P_6_, and P_8_ are good parents for most of the studied antioxidant enzymes, chlorophyll content, and relative water content. Additionally, the parents P_2_, P_5_, and P_6_ could be important sources of favorable genes for reducing plant height under both normal and stressed conditions. Furthermore, improving grain yield and its components could be achieved by exploiting P_3_, P_4_, and P_8_, which had constantly significant and positive GCA estimates under both conditions. Subsequently, these parents could inherit beneficial alleles to their progeny and improve grain yield under water-stressed conditions. Similarly, earlier reports stated the significance of employing parents with positive and high GCA estimates for improving grain yield and contributed traits under drought stress conditions [51,103,104]. Interestingly, the parent P_8_ exhibited favorable GCA estimates for grain yield and was also an excellent combiner for chlorophyll *b*, total chlorophyll content, relative water content, CAT activity, spike length, and 1000-grain weight. In consequence, it can be employed for breeding drought-tolerant and high-yielding genotypes in the wheat breeding program. The parents P_1_, P_2,_ and P_5_ were the best combiners for improving grain protein content, and P_1_ was the best for improving wet and dry glutens under both conditions. Subsequently, the valuable alleles of these genotypes could be employed in breeding programs for improving grain quality under well-watered and stressed conditions. Parental genotypes with desirable GCA effects for specific traits could be valuable for providing pure lines for breeding purposes.

Hybrids with significant SCA effects are great choices for selection. The result revealed that the crosses P_2_ × P_4_, P_3_ × P_5_, P_3_ × P_8,_ and P_6_ × P_7_ were identified as excellent combiners for developing high-yielding genotypes under stressed conditions. Out of the aforementioned crosses, three hybrids, P_2_ × P_4_, P_3_ × P_5_ and P_3_ × P_8_, exhibited desirable SCA and high grain yield. The relationship between high yield performance and desired SCA effects was also deduced by Semahegn, et al. [51] and Kamara, et al. [101]. Grain quality is considered by wheat breeders alongside grain yield to meet production demands and market needs [105]. Grain protein content affects the nutritional value and processing qualities of wheat [42,106]. The results demonstrated that the combinations P_1_ × P_4_, P_2_ × P_3_, P_2_ × P_5_, P_2_ × P_6_, P_4_ × P_7_, P_5_ × P_7_, P_5_ × P_8_, P_6_ × P_8_, and P_7_ × P_8_ were the most merit-specific combiners for protein content. Additionally, P_1_ × P_6_, P_2_ × P_3_, P_2_ × P_4_, P_3_ × P_7_, P_4_ × P_6_, P_5_ × P_6_, and P_5_ × P_7_ appeared as the best specific combiners for wet and dry gluten under drought and non-stressed conditions. These crosses could be used to develop new genotypes with higher protein and gluten levels for specific industrial applications, as suggested by Joshi, et al. [107]. The majority of specific crosses for grain yield and protein content included high×high and high×poor general combiners, which implies increasing the favorable alleles. Evidently, none of the assessed crosses displayed significant SCA effects for all evaluated traits. Nonetheless, P_3_ × P_8_ was a good combiner for chlorophyll *b*, total chlorophyll content, Fv/Fm, RWC, proline content, CAT, POD, spike length, number of grain per spike, 1000-grain weight, and grain yield per plant. Moreover, P_3_ × P_5_ and P_6_ × P_7_ were the best combiners for chlorophyll *b*, total chlorophyll content, RWC, spike length, number of grain per spike, grain yield per plant, and wet gluten content. Subsequently, these crosses could be effectively exploited in wheat breeding programs to ameliorate these traits under drought stress and normal conditions [108].

The cluster analysis classified the assessed genotypes into five groups (A–E) varied from highly tolerant to highly sensitive genotypes. The genotypes P_3_, P_8_, P_3_ × P_8_, P_4_ × P_8_, P_3_ × P_5_, P_6_ × P_7_, and P_5_ × P_6_ were classified to be drought-tolerant (Figure 5). These genotypes displayed higher agronomic performance compared with the sensitive ones. This superiority in agronomic performance was reflected by their superior efficiency in chlorophyll *a*, chlorophyll *b*, Fv/Fm, relative water content, proline content, and enzymatic antioxidants. Thereupon, these tolerant genotypes could be utilized in wheat breeding programs for boosting grain yield under water deficit conditions. Respectively, several previous reports applied tolerance indices and cluster analysis to classify wheat genotypes under drought stress conditions [109,110,111,112,113].

Both additive and non-additive gene actions were included in the inheritance of all evaluated traits under both treatments, as evidenced by the highly significant GCA and SCA effects for all the evaluated traits. However, the GCA/SCA ratio was greater than unity for the majority of the evaluated characteristics in both stressed and non-stressed conditions. This implies the preponderance of additive gene effects in controlling the inheritance of these traits. These findings coincide with those of Sinolinding and Chowdhry [114], Farshadfar, et al. [47], El-Maghraby, et al. [115], Rad, et al. [116], and Semahegn, et al. [51]. They demonstrated that the additive gene actions mainly contributed to the inheritance of several physiological and agronomic traits in wheat under drought and normal conditions. Otherwise, Mwadzingeni, et al. [103] manifested that non-additive gene action was more predominant in the inheritance of grain yield and other agronomic traits in a diallel study investigated under drought stress conditions. 

Understanding the associations among physiological, agronomic and quality traits could enhance breeding programs efficiency. The PC-biplot is a suitable statistical approach to assess the interrelationships among evaluated traits. The results displayed that grain yield was positively correlated with chlorophyll *a*, chlorophyll *b*, Fv/Fm, relative water content, and proline content under drought stress. Accordingly, selection for improving these physiological traits under water deficit conditions will result in improving the grain yield. These findings coincide with previous reports that reflected the importance of physiological attributes as indicators for grain yield under drought stress [117,118,119]. Strong positive relationships were detected between the plant height, the number of grains per spike, and the 1000-grain weight with grain yield, which implies their importance as valuable traits for indirect selection under drought stress [75,120]. On the other hand, grain yield was negatively associated with grain protein content, wet gluten content, and dry gluten content. Similarly, Ozturk and Aydin [41] Tari [121], Thungo, et al. [122], and Šíp, et al. [123] depicted a significant negative association between grain yield and grain protein, wet gluten content, and dry gluten content.

## 5. Conclusions

Water-deficit stress substantially reduced all assessed traits except proline content, antioxidant enzyme activities (CAT, POX, and SOD), grain protein content, wet gluten content, and dry gluten content, which were significantly increased. The parental genotypes P_3_ and P_8_ and their cross combination are proposed for breeding high-yielding and drought-tolerant wheat genotypes. Additionally, the parental genotypes P_1_, P_2_, and P_5_, as well as the hybrid combinations P_1_ × P_4_, P_2_ × P_3_, P_2_ × P_5_, P_2_ × P_6_, P_4_ × P_7_, P_5_ × P_7_, P_5_ × P_8_, P_6_ × P_8_, and P_7_ × P_8_, were the most promising genotypes for improving grain quality traits under drought-stressed conditions. The tolerance indices and cluster analysis provide valuable information on classifying the genotypes based on their tolerance to drought stress. Chlorophyll a, chlorophyll b, photosynthetic efficiency, proline content, and relative water content displayed highly positive associations with grain yield and its contributing traits under drought stress. These findings suggest the importance of these traits for indirect selection under water deficit conditions.

## Figures and Tables

**Figure 1 plants-11-00952-f001:**
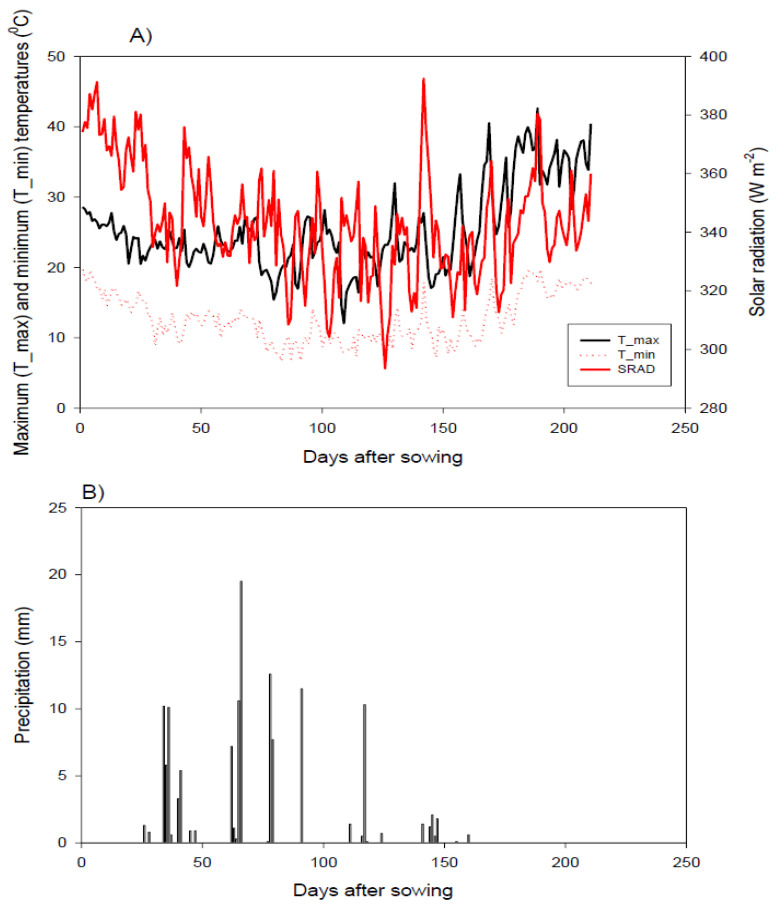
Daily minimum and maximum temperatures, as well as solar radiation (**A**) and precipitation (**B**), at the experimental site.

**Figure 2 plants-11-00952-f002:**
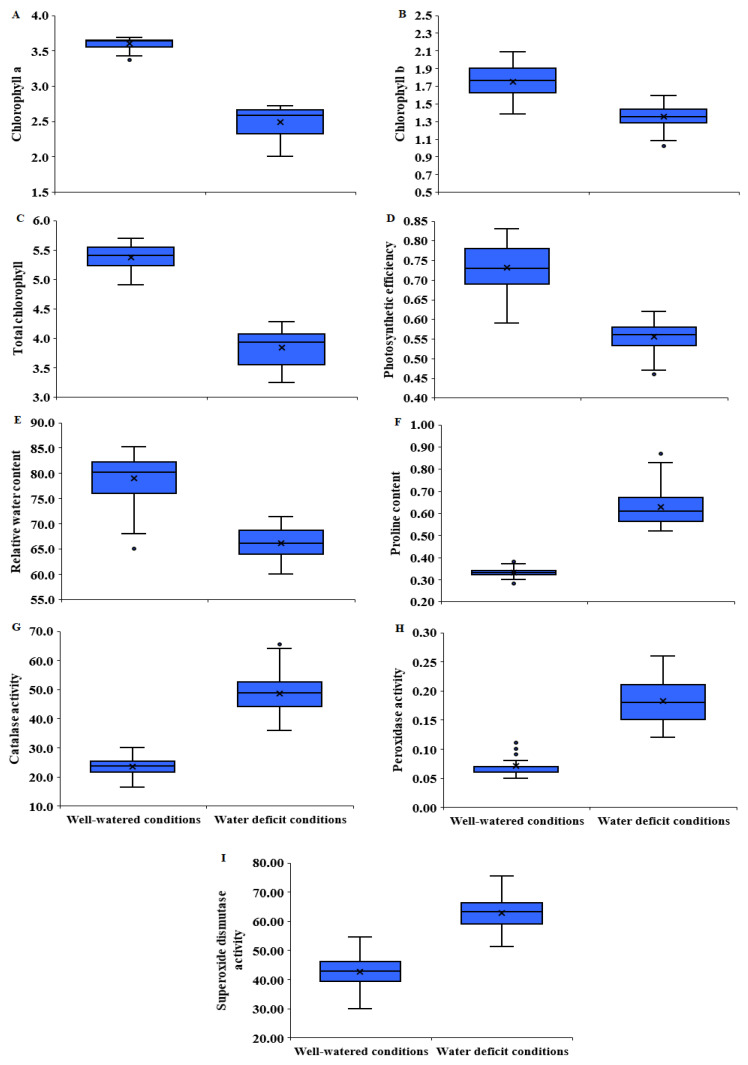
Boxplots with minimum, median, mean, and maximum values for chlorophyll *a* (**A**), chlorophyll *b* (**B**), total chlorophyll content (**C**), photosynthetic efficiency (**D**), relative water content (**E**), proline content (**F**), catalase activity (**G**), peroxidase activity (**H**), and superoxide dismutase activity (**I**).

**Figure 3 plants-11-00952-f003:**
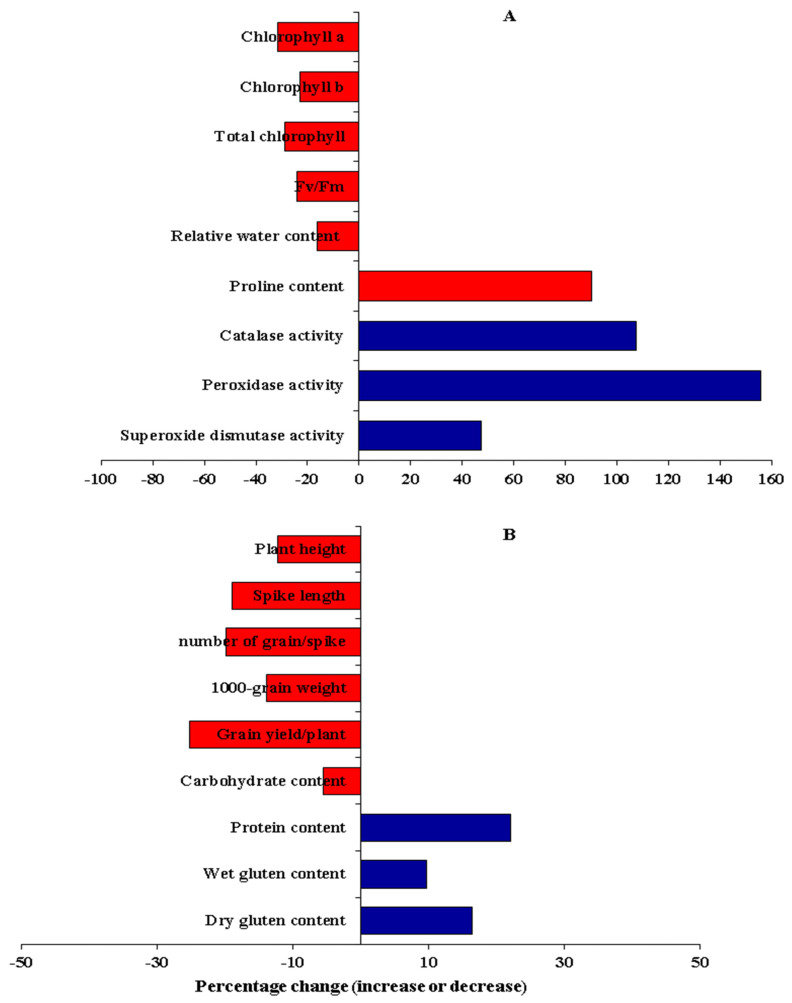
Percentage change (increase or decrease) in physiological and biochemical traits (**A**) and agronomic and grain quality traits (**B**) exposed to drought stress compared with well-watered wheat plants.

**Figure 4 plants-11-00952-f004:**
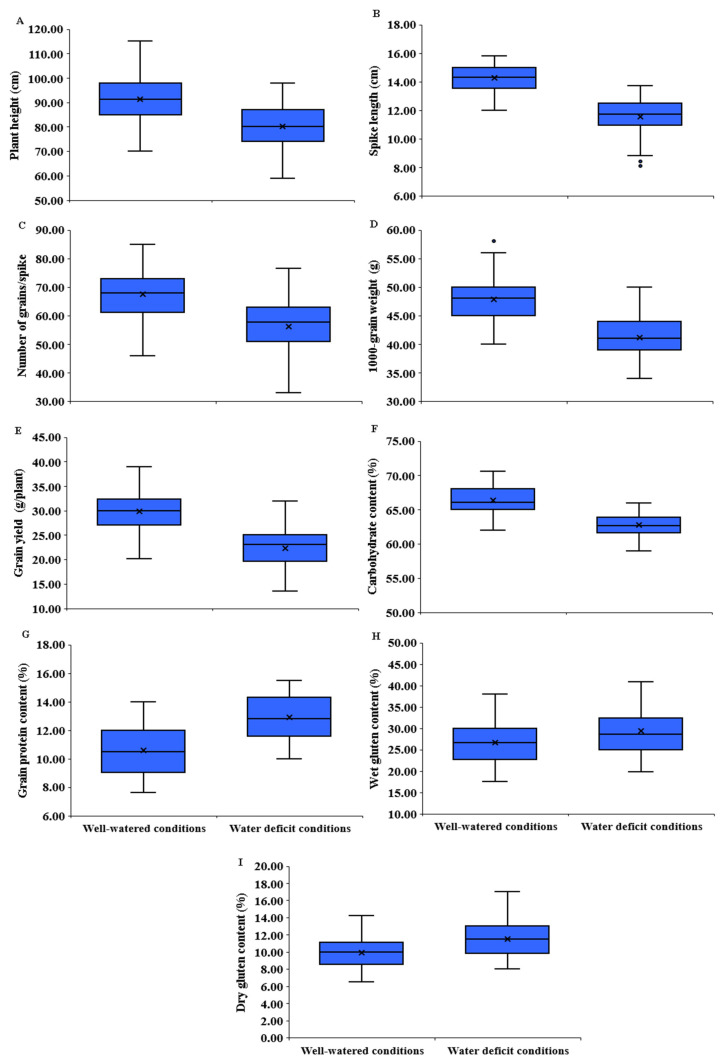
Boxplots with minimum, median, mean, and maximum values for plant height (**A**), spike length (**B**), number of grains per spike (**C**), 1000-grain weight (**D**), grain yield per plant (**E**), carbohydrate content (**F**), (**G**) protein content, (**H**) wet gluten content, and (**I**) dry gluten content.

**Figure 5 plants-11-00952-f005:**
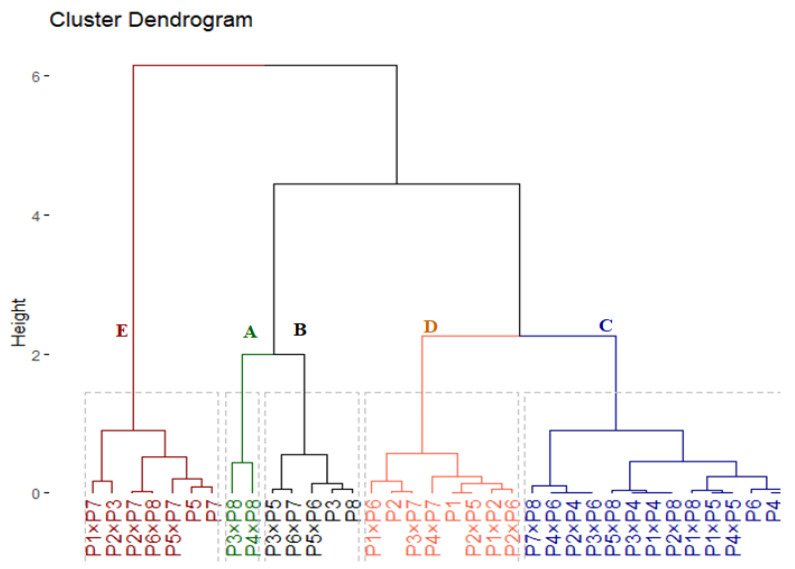
Dendrogram of the phenotypic distances among eight wheat genotypes and their 28 F_1_s based on 4 drought tolerance indices (GM, MP, YI, and STI).

**Figure 6 plants-11-00952-f006:**
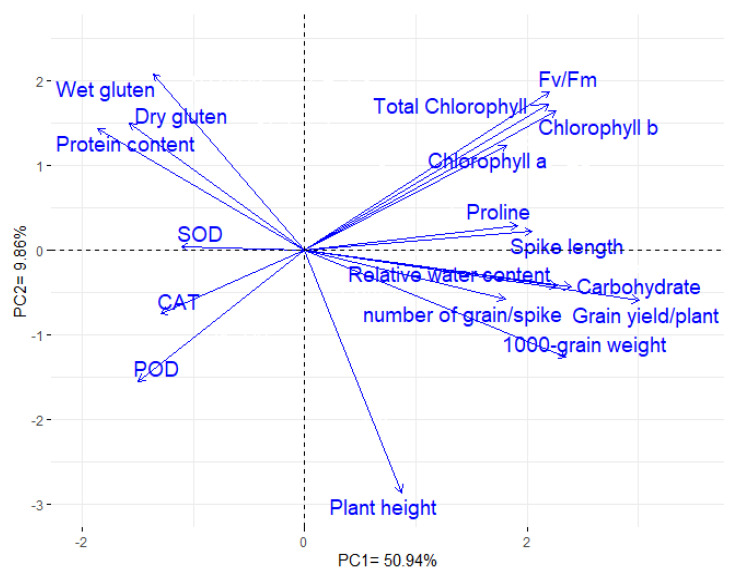
PC-biplot presents the relationship among the evaluated physiological, agronomic, and quality traits.

**Table 1 plants-11-00952-t001:** Mean squares from ordinary and combining ability analysis for all investigated traits under well-watered and drought stress conditions.

**Source of Variance**	**DF**	**Chlorophyll *a* (mg g^−1^ FW)**		**Chlorophyll *b* (mg g^−1^ FW)**		**Total Chlorophyll (mg g^−1^ FW)**		**Fv/Fm**		**RWC (%)**		**Proline (mg g^−1^ FW)**	
		**Well-Watered**	**Water Deficit**	**Well-Watered**	**Water Deficit**	**Well-Watered**	**Water Deficit**	**Well-Watered**	**Water Deficit**	**Well-Watered**	**Water Deficit**	**Well-Watered**	**Water Deficit**
Genotypes (G)	35	0.01 **	0.12 **	0.08 **	0.06 **	0.11 **	0.25 **	0.007 **	0.003 **	54.89 **	26.25 **	0.0008 **	0.015 **
Parents (P)	7	0.01 **	0.06 **	0.16 **	0.05 **	0.15 **	0.10 **	0.003 **	0.003 **	62.97 **	20.05 **	0.0008 **	0.018 **
F_1_ Crosses (C)	27	0.01 **	0.14 **	0.07 **	0.06 **	0.10 **	0.28 **	0.008 **	0.003 **	52.00 **	28.74 **	0.0008 **	0.014 **
P vs. C	1	0.003	0.12 **	0.02 **	0.21 **	0.03 **	0.64 **	0.017 **	0.007 **	76.54 **	2.47 **	0.00004	0.025 **
Error	70	0.002	0.001	0.0002	0.0004	0.002	0.001	0.001 **	0.001 **	0.52	0.35	0.0001	0.002
GCA	7	0.036 **	0.0038 **	0.03 **	0.03 **	0.04 **	0.07 **	0.003 **	0.001 *	29.13 **	12.44 **	0.00023 **	0.007 **
SCA	28	0.043 **	0.0036 **	0.03 **	0.02 **	0.04 **	0.09 **	0.002 **	0.001 **	15.59 **	7.83 **	0.0003 **	0.005 **
Error term	70	0.001	0.0004	0.0001	0.0001	0.001	0.0005	0.0003	0.0002	0.17	0.12	0.00004	0.001
GCA/SCA		0.85	1.06	1.19	1.59	1.03	0.75	1.32	0.62	1.87	1.59	0.87	1.37
		**CAT (Unit mg^−1^ Protein)**		**POD** **(Unit mg^−1^ Protein)**		**SOD** **(Unit mg^−1^ Protein)**		**Plant Height (cm)**		**Spike Length (cm)**		**Number of Grains/Spike**	
		**Well-Watered**	**Water Deficit**	**Well-Watered**	**Water Deficit**	**Well-Watered**	**Water Deficit**	**Well-Watered**	**Water Deficit**	**Well-Watered**	**Water Deficit**	**Well-Watered**	**Water Deficit**
Genotypes (G)	35	19.01 **	114.8 **	0.0004 **	0.003 **	58.55 **	62.58 **	202.85 **	210.2 **	2.22 **	2.98 **	215.6 **	235.4 **
Parents (P)	7	14.49 **	44.22 **	0.0012 **	0.002 **	47.49 **	24.13 **	180.27 **	157.5 **	2.79 **	3.75 **	211.2 **	157.3 **
F_1_ Crosses (C)	27	18.22 **	135.8 **	0.0002 **	0.002 **	63.33 **	74.67 **	149.49 **	163.0 **	2.04 **	2.77 **	222.8 **	263.1 **
P vs. C	1	72.04 **	42.64 **	0.0023 **	0.039 **	6.89	5.30	1801.58 **	1851.9 **	3.05 **	3.07 *	52.00 *	33.56 *
Error	70	2.91	3.95	0.00001	0.0001	4.96	5.68	14.22	12.49	0.24	0.49	8.92	7.93
GCA	7	2.57 *	59.08 **	0.0001 **	0.0009 **	22.59 **	17.46 **	97.95 **	86.91 **	1.42 **	1.80 **	151.4 **	154.9 **
SCA	28	7.28 **	33.08 **	0.0001 **	0.001 **	18.75 **	21.71 **	60.03 **	65.85 **	0.57 **	0.79 **	51.98 **	59.33 **
Error term	70	0.97	1.32	0.000002	0.00004	1.65	1.89	4.74	4.16	0.08	0.16	2.97	2.64
GCA/SCA		0.35	1.79	1.02	0.93	1.21	0.80	1.63	1.32	2.50	2.28	2.91	2.61
		**1000-Grain Weight (g)**		**Grain Yield/Plant (g)**		**Carbohydrates Content (%)**		**Grain Protein Content (%)**		**Wet Gluten (%)**		**Dry Gluten (%)**	
		**Well-Watered**	**Water Deficit**	**Well-Watered**	**Water Deficit**	**Well-Watered**	**Water Deficit**	**Well-Watered**	**Water Deficit**	**Well-Watered**	**Water Deficit**	**Well-Watered**	**Water Deficit**
Genotypes (G)	35	43.61 **	28.12 **	38.99 **	37.75 **	13.66 **	7.69 **	7.71 **	6.57 **	74.36 **	74.29 **	10.44 **	11.11 **
Parents (P)	7	38.36 **	27.02 **	26.49 **	29.95 **	8.91 **	6.12 **	7.92 **	6.24 **	63.96 **	68.85 **	17.73 **	14.35 **
F_1_ Crosses (C)	27	46.59 **	28.58 **	43.04 **	40.62 **	13.59 **	7.98 **	7.60 **	6.89 **	76.27 **	73.83 **	8.94 **	10.64 **
P vs. C	1	0.003	23.38 **	16.95 *	14.92 *	48.87 **	10.79 **	9.20 **	0.36	95.36 **	124.9 **	0.12	1.41
Error	70	3.29	2.70	4.13	3.68	0.38	0.43	0.17	0.14	0.63	0.52	0.37	0.43
GCA	7	20.39 **	10.35 **	31.33 **	26.11 **	3.86 **	1.93 **	2.86 **	3.35 **	37.77 **	38.12 **	6.38 **	5.74 **
SCA	28	13.07 **	9.13 **	8.41 **	9.20 **	4.73 **	2.72 **	2.50 **	1.90 **	21.54 **	21.42 **	2.76 **	3.20 **
Error term	70	1.10	0.90	1.38	1.23	0.13	0.14	0.06	0.05	0.21	0.17	0.12	0.14
GCA/SCA	35	1.56	1.13	3.72	2.84	0.82	0.71	1.10	1.76	1.75	1.78	2.31	1.80

DF is degree of freedom; * and ** indicate *p* < 0.05 and 0.01, respectively.

**Table 2 plants-11-00952-t002:** General combining ability estimates (GCA) of the eight parents for all assessed traits under well-watered and drought stress conditions.

**Parent**	**Chlorophyll *a* (mg g^−1^ FW)**		**Chlorophyll *b* (mg g^−1^ FW)**		**Total Chlorophyll (mg g^−1^ FW)**		**Fv/Fm**		**RWC (%)**		**Proline (mg g^−1^ FW)**	
	**Well-Watered**	**Water Deficit**	**Well-Watered**	**Water Deficit**	**Well-Watered**	**Water Deficit**	**Well-Watered**	**Water Deficit**	**Well-Watered**	**Water Deficit**	**Well-Watered**	**Water Deficit**
P_1_	0.003	0.05 **	−0.07 **	−0.10 **	−0.07 **	−0.05 **	−0.003	0.01 *	−0.10	−2.18 **	−0.001	−0.011
P_2_	0.01 *	0.02 **	−0.07 **	0.04 **	−0.06 **	0.06 **	0.014 **	0.007	−1.93 **	−0.46 **	−0.005 **	−0.04 **
P_3_	0.01 *	−0.01 *	0.08 **	0.08 **	0.10 **	0.07 **	0.0002	0.009 *	3.01 **	1.33 **	0.009 **	0.04 **
P_4_	0.03 **	−0.09 **	−0.01 **	0.0005	0.01	−0.09 **	−0.025 **	−0.011 *	−0.08	0.55 **	−0.003	−0.01 *
P_5_	−0.01 *	−0.02 **	−0.03 **	−0.006	−0.04 **	−0.03 **	−0.011 *	−0.009 *	1.00 **	−0.27 **	−0.004 *	−0.01 *
P_6_	−0.02 **	0.04 **	−0.003	−0.01 **	−0.02 **	0.03 **	−0.001	−0.002	0.25 *	0.24 *	−0.001	0.003
P_7_	−0.03 **	−0.07 **	0.04 **	−0.04 **	0.01	−0.11 **	−0.006	−0.002	−2.52 **	−0.38 **	0.00001	0.03 **
P_8_	−0.01 *	0.08 **	0.06 **	0.03 **	0.07 **	0.12 **	0.032 **	−0.002	0.36 **	1.17 **	0.004 *	−0.01 *
LSD (gi) _0_._05_	0.01	0.01	0.01	0.01	0.02	0.01	0.011	0.009	0.24	0.20	0.004	0.01
LSD (gi) _0_._01_	0.02	0.02	0.01	0.01	0.02	0.02	0.014	0.012	0.32	0.27	0.005	0.02
	**CAT (Unit mg^−1^ Protein)**		**POD** **(Unit mg^−1^ Protein)**		**SOD** **(Unit mg^−1^ Protein)**		**Plant Height (cm)**		**Spike Length (cm)**		**Number of Grains/Spike**	
	**Well-Watered**	**Water Deficit**	**Well-Watered**	**Water Deficit**	**Well-Watered**	**Water Deficit**	**Well-Watered**	**Water Deficit**	**Well-Watered**	**Water Deficit**	**Well-Watered**	**Water Deficit**
P_1_	−0.18	−1.78 **	−0.004 **	−0.015 **	0.99 *	2.09 **	−1.26	−0.68	−0.37 **	−0.21	−1.40 **	−2.07 **
P_2_	0.04	0.84 *	−0.005 **	0.012 **	−1.29 **	−0.89 *	−4.85 **	−5.10 **	0.24 **	0.20	4.18 **	2.68 **
P_3_	−0.35	−2.23 **	0.002 **	0.001	0.45	0.92 *	2.13 **	1.96 **	−0.28 **	−0.63 **	2.82 **	0.38
P_4_	−0.31	−3.16 **	0.0001	−0.010 **	−0.90 *	−2.05 **	1.38 *	1.48 *	0.42 **	0.46 **	3.58 **	5.34 **
P_5_	−0.23	0.39	−0.001	−0.001	−1.90 **	−0.70	−1.33 *	−1.33 *	−0.07	−0.05	−3.62 **	−2.36 **
P_6_	1.19 **	−0.35	0.006 **	0.013 **	0.48	0.59	−1.83 **	−1.30 *	−0.51 **	−0.52 **	0.76	2.48 **
P_7_	−0.25	4.23 **	−0.003 **	0.003	−0.64	−0.75	0.11	−0.02	0.04	0.32 **	−7.17 **	−7.46 **
P_8_	0.09	2.07 **	0.004 **	−0.003	2.81 **	0.78	5.65 **	4.99 **	0.53 **	0.43 **	0.85	1.01 *
LSD (gi) _0_._05_	0.58	0.68	0.001	0.004	0.76	0.81	1.28	1.20	0.17	0.24	1.02	0.96
LSD (gi) _0_._01_	0.77	0.90	0.001	0.005	1.00	1.07	1.70	1.59	0.22	0.31	1.35	1.27
	**1000-Grain Weight (g)**		**Grain Yield/Plant (g)**		**Carbohydrate Content (%)**		**Grain Protein Content (%)**		**Wet Gluten (%)**		**Dry Gluten (%)**	
	**Well-Watered**	**Water Deficit**	**Well-Watered**	**Water Deficit**	**Well-Watered**	**Water Deficit**	**Well-Watered**	**Water Deficit**	**Well-Watered**	**Water Deficit**	**Well-Watered**	**Water Deficit**
P_1_	−1.42 **	−0.38	0.92 **	−1.02 **	−0.01	−0.60 **	0.23 **	0.47 **	2.52 **	1.86 **	0.66 **	1.42 **
P_2_	−1.62 **	−1.61 **	−0.83 *	−1.48 **	−0.25 *	−0.22 *	0.75 **	0.50 **	1.19 **	1.82 **	0.10	−0.03
P_3_	0.82 *	0.56 *	1.58 **	1.49 **	0.64 **	0.60 **	−0.80 **	−0.41 **	−2.56 **	−2.74 **	−0.75 **	−0.58 **
P_4_	1.32 **	0.23	1.00 **	1.23 **	1.10 **	0.45 **	−0.55 **	0.15 *	−1.29 **	−1.80 **	0.27 *	−0.11
P_5_	0.15	−0.34	−0.57	−0.87 **	−0.53 **	−0.05	0.33 **	0.70 **	−1.52 **	−1.10 **	1.17 **	0.67 **
P_6_	−0.62 *	−0.67 *	−1.22 **	0.69 *	−0.84 **	−0.56 **	0.16 *	−0.53 **	−1.46 **	−0.98 **	−0.68 **	−0.69 **
P_7_	0.42	0.36	−3.14 **	−2.27 **	−0.17	0.20	0.32 **	0.07	2.32 **	2.62 **	0.40 **	0.16
P_8_	2.58 **	1.86 **	2.28 **	2.23 **	0.05	0.18	−0.45 **	−0.95 **	0.79 **	0.32 *	−1.17 **	−0.83 **
LSD (gi) _0_._05_	0.62	0.56	0.69	0.65	0.21	0.22	0.14	0.13	0.27	0.24	0.21	0.22
LSD (gi) _0_._01_	0.82	0.74	0.92	0.86	0.28	0.29	0.19	0.17	0.36	0.32	0.27	0.30

* and ** indicate *p* < 0.05 and 0.01, respectively.

**Table 3 plants-11-00952-t003:** Specific combining ability effects (SCA) of 28 F_1_ cross combinations for all studied traits under well-watered and drought stress conditions.

**Cross**	**Chlorophyll *a*** **(mg g^−1^ FW)**		**Chlorophyll *b*** **(mg g^−1^ FW)**		**Total Chlorophyll** **(mg g^−1^ FW)**		**Fv/Fm**		**RWC (%)**		**Proline** **(mg g^−1^ FW)**		**CAT** **(Unit mg^−1^ Protein)**		**POD** **(Unit mg^−1^ Protein)**		**SOD** **(Unit mg^−1^ Protein)**	
	**Well-** **Watered**	**Water Deficit**	**Well-** **Watered**	**Water Deficit**	**Well-** **Watered**	**Water Deficit**	**Well-** **Watered**	**Water Deficit**	**Well-** **Watered**	**Water Deficit**	**Well-** **Watered**	**Water Deficit**	**Well-** **Watered**	**Water Deficit**	**Well-** **Watered**	**Water Deficit**	**Well-** **Watered**	**Water Deficit**
P_1_ × P_2_	0.02	0.03	−0.11 **	−0.10 **	−0.09 **	−0.06 **	0.03	−0.01	−0.84 *	−2.69 **	0.01	0.04 *	−1.14	2.58 *	0.008 **	0.03 **	−3.72 **	−5.87 **
P_1_ × P_3_	−0.003	0.15 **	−0.35 **	−0.04 **	−0.36 **	0.11 **	−0.04 *	0.003	−0.88 *	−2.08 **	−0.03 **	0.003	0.25	1.64	−0.005 **	−0.03 **	0.96	0.34
P_1_ × P_4_	0.01	0.22 **	−0.01	0.11 **	−0.01	0.32 **	−0.08 **	0.004	1.91 **	1.13 **	0.01 *	−0.05 *	2.22 *	−0.24	0.003 *	−0.03 **	−2.39 *	−2.49 *
P_1_ × P_5_	−0.09 **	0.17 **	0.12 **	0.10 **	0.03	0.27 **	0.05 **	0.02	0.21	−2.65 **	0.002	−0.002	0.13	−9.00 **	0.001	−0.01	1.48	0.30
P_1_ × P_6_	0.04 *	0.09 **	0.09 **	0.0001	0.12 **	0.08 **	−0.003	0.03 *	3.62 **	2.84 **	−0.01	0.08 **	2.18 *	2.72 *	−0.012 **	−0.02 *	0.18	1.70
P_1_ × P_7_	0.02	−0.33 **	−0.08 **	0.09 **	−0.07 **	−0.24 **	−0.02	−0.03 *	0.66	−0.21	−0.01	0.01	−3.49 **	0.19	−0.004 **	0.03 **	−0.22	4.95 **
P_1_ × P_8_	0.0003	0.05 **	0.13 **	0.09 **	0.13 **	0.14 **	0.04 *	0.00	−6.41 **	−3.01 **	0.004	−0.02	−2.81 **	−0.64	−0.004 **	−0.01	4.12 **	5.17 **
P_2_ × P_3_	−0.002	−0.40 **	−0.14 **	−0.14 **	−0.14 **	−0.54 **	−0.03	−0.07 **	1.23 **	−5.97 **	−0.02 **	−0.08 **	2.50 **	8.53 **	0.003 *	0.02 *	6.19 **	4.10 **
P_2_ × P_4_	0.0003	0.15 **	0.27 **	0.13 **	0.27 **	0.28 **	0.07 **	0.01	2.22 **	1.88 **	0.02 **	0.02	0.42	−8.59 **	0.015 **	−0.03 **	2.49 *	4.32 **
P_2_ × P_5_	0.02	0.10 **	0.20 **	0.17 **	0.22 **	0.27 **	−0.03	0.02	−1.82 **	0.83 **	0.01	−0.01	0.81	−1.49	0.0001	−0.02 **	1.94	1.61
P_2_ × P_6_	0.03	0.11 **	−0.07 **	0.07 **	−0.04	0.18 **	0.004	0.01	−6.41 **	1.09 **	−0.02 **	0.01	−1.61	−3.80 **	0.002	0.00	−7.97 **	−7.97 **
P_2_ × P_7_	−0.09 **	0.23 **	−0.10 **	0.004	−0.20 **	0.23 **	−0.10 **	0.01	6.07 **	0.88 **	−0.01 *	−0.04	2.83 **	−4.93 **	0.011 **	−0.03 **	3.14 **	3.07 *
P_2_ × P_8_	0.001	0.06 **	0.15 **	0.03 **	0.15 **	0.09 **	0.01	0.02	4.34 **	4.40 **	−0.01	0.003	−1.50	6.77 **	−0.008 **	−0.05 **	0.20	4.36 **
P_3_ × P_4_	−0.02	0.23 **	0.02 **	0.16 **	0.01	0.39 **	−0.04 *	0.02	−0.11	−0.83 **	0.02 **	−0.07 **	1.72	2.47 *	−0.004 **	−0.02 **	−10.60 **	−8.15 **
P_3_ × P_5_	0.03	0.21 **	0.15 **	0.14 **	0.18 **	0.35 **	0.08 **	0.03 *	0.95 *	1.37 **	0.03 **	0.12 **	−4.44 **	−5.35 **	−0.007 **	0.01	2.64 *	−5.10 **
P_3_ × P_6_	0.04 *	0.14 **	−0.01	−0.05 **	0.03	0.08 **	−0.02	−0.02	0.16	1.10 **	0.02 **	−0.01	−1.81 *	−3.38 **	−0.016 **	−0.03 **	2.74 *	7.26 **
P_3_ × P_7_	0.05 *	−0.03	−0.05 **	0.03 **	0.002	0.001	0.01	0.001	4.37 **	3.08 **	−0.03 **	0.04 *	−2.40 **	−7.97 **	−0.001	−0.03 **	−1.43	−0.10
P_3_ × P_8_	0.01	0.13 **	0.09 **	0.10 **	0.10 **	0.24 **	0.04 *	0.03 *	1.42 **	2.02 **	0.02 **	0.10 **	2.55 **	5.64 **	0.013 **	0.01 *	1.63	2.75 *
P_4_ × P_5_	0.02	−0.21 **	0.12 **	−0.04 **	0.13 **	−0.25 **	−0.07 **	0.02	4.98 **	3.47 **	−0.01 *	0.001	−3.75 **	4.58 **	−0.008 **	−0.04 **	0.13	2.50 *
P_4_ × P_6_	0.01	−0.27 **	0.18 **	0.02 *	0.19 **	−0.25 **	0.02	0.02	−3.17 **	−2.63 **	−0.002	0.12 **	−4.17 **	−5.67 **	−0.008 **	0.02 **	2.66 *	−5.42 **
P_4_ × P_7_	0.02	−0.21 **	0.15 **	−0.17 **	0.18 **	−0.38 **	0.01	−0.06 **	−0.73	−1.44 **	−0.004	−0.11 **	2.29 *	1.61	−0.002	0.01	−2.04	3.34 **
P_4_ × P_8_	0.005	0.20 **	−0.06 **	−0.06 **	−0.06 *	0.14 **	−0.02	0.06 **	2.44 **	1.33 **	0.01	0.09 **	−0.05	−5.76 **	−0.003 *	−0.02 *	0.51	−3.44 **
P_5_ × P_6_	0.05 *	0.11 **	−0.001	0.17 **	0.05 *	0.28 **	0.01	0.03 *	−1.74 **	3.69 **	−0.002	0.05 *	0.21	−8.23 **	−0.006 **	−0.04 **	2.68 *	1.16
P_5_ × P_7_	−0.18 **	−0.23 **	−0.16 **	−0.19 **	−0.34 **	−0.42 **	−0.03	0.02	−10.18 **	0.57	−0.01 *	−0.09 **	0.63	7.02 **	0.01 **	0.02 **	−0.99	5.58 **
P_5_ × P_8_	0.03	−0.11 **	−0.18 **	−0.30 **	−0.14 **	−0.40 **	−0.02	−0.06 **	2.33 **	−2.13 **	−0.01	0.07 **	3.35 **	0.39	−0.009 **	−0.02 *	−4.45 **	−3.84 **
P_6_ × P_7_	0.10 **	0.18 **	0.13 **	0.18 **	0.23 **	0.36 **	0.01	0.02	2.85 **	1.70 **	0.01	0.02	−1.81 *	11.90 **	0.001	−0.02 **	6.03 **	−5.38 **
P_6_ × P_8_	−0.12 **	−0.37 **	−0.17 **	0.01	−0.28 **	−0.36 **	−0.04 **	−0.03 *	2.04 **	−0.61	0.01	−0.08 **	−0.11	0.09	−0.011 **	0.02 **	5.62 **	4.09 **
P_7_ × P_8_	0.02	0.09 **	−0.08 **	0.14 **	−0.06 *	0.23 **	−0.02	0.01	3.07 **	−4.87 **	0.002	0.01	−5.22 **	−0.49	−0.026 **	−0.03 **	−7.73 **	−5.55 **
LSD Sij _0_._05_	0.04	0.04	0.02	0.02	0.05	0.04	0.03	0.03	0.75	0.62	0.01	0.04	1.78	2.07	0.003	0.01	2.32	2.48
LSD Sij _0_._01_	0.06	0.05	0.02	0.03	0.06	0.05	0.04	0.04	0.99	0.82	0.01	0.05	2.36	2.75	0.004	0.02	3.08	3.29
**Cross**	**Plant Height (cm)**		**Spike Length (cm)**		**Number of Grains/Spike**		**1000-Grain Weight (g)**		**Grain Yield/Plant (g)**		**Carbohydrate** **Content (%)**		**Grain Protein** **Content (%)**		**Wet Gluten (%)**		**Dry Gluten (%)**	
	**Well-** **Watered**	**Water Deficit**	**Well-** **Watered**	**Water Deficit**	**Well-** **Watered**	**Water Deficit**	**Well-** **Watered**	**Water Deficit**	**Well-** **Watered**	**Water Deficit**	**Well-** **Watered**	**Water Deficit**	**Well-** **Watered**	**Water Deficit**	**Well-** **Watered**	**Water Deficit**	**Well-** **Watered**	**Water Deficit**
P_1_ × P_2_	4.22 *	4.39 *	−0.20	−0.27	−12.81 **	−9.46 **	1.61	1.20	2.38 *	1.01	4.29 **	3.08 **	−1.67 **	−2.72 **	−8.28 **	−8.95 **	−1.88 **	−1.81 **
P_1_ × P_3_	−6.07 **	−6.06 **	−0.72 **	−0.31	0.44	7.54 **	1.47	2.70 **	−1.87	0.18	2.22 **	0.63	−0.84 **	−0.29	−4.27 **	−4.19 **	0.70 *	1.65 **
P_1_ × P_4_	−11.64 **	−10.29 **	−0.19	−0.44	2.70	0.56	−4.99 **	−3.97 **	0.36	0.84	−2.93 **	−0.86 *	2.40 **	1.67 **	−3.72 **	−3.72 **	0.81 *	0.67
P_1_ × P_5_	1.97	1.95	−0.52 *	−0.52	0.39	0.44	0.84	0.26	3.17 **	1.93	2.59 **	1.01 **	−2.40 **	−1.14 **	0.32	−0.17	1.61 **	1.09 **
P_1_ × P_6_	−8.85 **	−8.13 **	0.08	0.39	2.50	4.13 **	−1.39	−2.07 *	−1.33	−0.16	0.11	0.15	0.37	1.44 **	2.95 **	1.84 **	1.30 **	1.92 **
P_1_ × P_7_	−8.78 **	−9.73 **	−0.60 *	−0.59	4.14 **	−4.65 **	−5.43 **	−4.10 **	−2.66 *	−3.00 **	0.17	−0.71 *	−0.42	1.13 **	6.22 **	5.84 **	0.51	0.50
P_1_ × P_8_	5.26 **	6.19 **	0.74 **	0.60	14.43 **	4.43 **	2.41 *	0.73	2.32 *	−1.03	1.59 **	−1.62 **	−0.93 **	−1.35 **	6.15 **	6.68 **	0.22	0.70 *
P_2_ × P_3_	−6.61 **	−10.79 **	−1.17 **	−1.76 **	2.80	−4.90 **	−2.66 **	−5.07 **	−5.03 **	−6.89 **	−0.46	−2.08 **	0.52 *	1.34 **	9.36 **	9.66 **	2.76 **	1.43 **
P_2_ × P_4_	9.78 **	5.13 **	0.21	0.35	6.15 **	7.56 **	3.54 **	−0.74	4.67 **	2.30 *	0.45	−0.24	0.15	0.74 **	2.34 **	1.24 **	0.91 **	1.18 **
P_2_ × P_5_	1.15	6.68 **	0.08	0.51	−3.68 *	−2.53	9.04 **	4.83 **	1.25	0.39	0.01	−0.33	1.20 **	1.32 **	−4.71 **	−5.99 **	−4.20 **	−3.71 **
P_2_ × P_6_	−8.08 **	−8.77 **	−0.49	0.51	8.94 **	−5.54 **	−2.53 **	1.16	−1.65	−0.90	0.89 **	0.04	1.00 **	1.35 **	1.27 **	4.56 **	−1.31 **	−1.01 **
P_2_ × P_7_	−3.95 *	−4.04 *	0.67 *	−0.09	3.57 *	11.60 **	−0.23	1.46	−2.38 *	−0.21	−2.28 **	−0.35	0.30	−0.95 **	−7.83 **	−5.38 **	−1.06 **	−1.78 **
P_2_ × P_8_	−5.60 **	−7.33 **	0.60 *	0.54	−6.17 **	−4.11 **	−3.06 **	0.30	0.40	0.29	2.83 **	2.81 **	−3.03 **	−1.43 **	6.29 **	5.86 **	−0.62	−0.79 *
P_3_ × P_4_	−11.58 **	−9.27 **	−0.61 *	−0.91 *	−6.73 **	−15.22 **	1.07	3.10 **	−4.03 **	−1.66	0.40	0.18	−0.67 **	0.19	1.38 **	1.00 **	−0.82 *	−0.22
P_3_ × P_5_	7.51 **	3.20	1.19 **	1.48 **	8.77 **	7.14 **	0.24	−0.34	3.06 **	3.77 **	−0.14	0.75 *	−0.26	1.11 **	0.94 *	0.57	−2.86 **	−3.07 **
P_3_ × P_6_	−8.45 **	−6.67 **	1.53 **	−0.06	−6.84 **	−5.67 **	−2.66 **	−1.67	0.38	−0.13	−0.71 *	−0.01	0.21	−1.27 **	−4.82 **	−3.38 **	−1.59 **	−1.73 **
P_3_ × P_7_	−0.77	0.90	0.14	0.69	−2.30	−5.94 **	2.64 **	4.30 **	3.53 **	−0.28	2.35 **	2.20 **	−2.19 **	−2.02 **	1.90 **	1.25 **	1.96 **	2.59 **
P_3_ × P_8_	−2.13	2.22	0.72 **	1.32 **	10.30 **	10.10 **	4.81 **	4.80 **	3.93 **	5.00 **	1.61 **	0.86 *	0.16	−0.34	−2.04 **	−1.88 **	−0.12	−0.28
P_4_ × P_5_	−5.25 **	−10.98 **	0.36	0.47	4.13 *	4.97 **	−0.89	3.66 **	1.43	−0.31	2.27 **	0.76 *	−2.67 **	−1.37 **	−4.78 **	−3.04 **	−1.00 **	−0.67
P_4_ × P_6_	−2.93	−3.01	0.46	0.81 *	−2.30	2.17	−2.46 *	0.33	−0.05	0.13	0.79 *	2.07 **	0.25	−1.00 **	5.08 **	4.34 **	1.43 **	1.35 **
P_4_ × P_7_	2.15	7.91 **	0.52 *	0.43	0.19	−0.10	1.17	1.63	−0.97	−1.38	1.99 **	−2.02 **	1.01 **	1.07 **	−1.50 **	−2.10 **	−0.56	−0.96 **
P_4_ × P_8_	−1.50	−1.13	0.14	0.29	−3.08	3.14 *	−2.66 **	−1.54	0.52	3.70 **	0.09	2.00 **	−0.17	−0.32	0.40	0.61	−0.25	−1.17 **
P_5_ × P_6_	−0.82	3.62	1.12 **	1.06 **	9.03 **	7.03 **	−0.29	1.90 *	1.77	3.69 **	−1.99 **	−0.86 *	−2.72 **	−2.28 **	3.41 **	3.94 **	1.62 **	1.47 **
P_5_ × P_7_	−7.14 **	−8.82 **	−0.56 *	−0.21	−9.24 **	−12.91 **	−6.66 **	−4.47 **	−4.89 **	−2.48 *	−2.00 **	−0.45	0.79 **	0.79 **	5.02 **	5.21 **	0.86 **	0.99 **
P_5_ × P_8_	−2.68	2.04	−0.07	−0.66	2.07	6.78 **	−2.83 **	−0.97	1.77	−0.21	−0.82 *	−1.13 **	2.18 **	1.73 **	0.17	0.01	0.24	−0.75 *
P_6_ × P_7_	8.37 **	8.15 **	0.59 *	1.29 **	4.99 **	6.40 **	2.77 **	3.86 **	3.62 **	6.16 **	1.11 **	2.99 **	−0.05	0.28	1.30 **	1.08 **	−0.55	−1.60 **
P_6_ × P_8_	0.16	−4.52 *	−1.28 **	−1.86 **	−13.64 **	−7.86 **	2.94 **	−1.30	−4.21 **	−6.94 **	−0.84 *	−1.22 **	1.55 **	1.80 **	0.00	−0.61	0.86 **	1.51 **
P_7_ × P_8_	1.12	−4.80 *	−0.20	−0.54	−8.38 **	−13.41 **	4.24 **	−3.00 **	0.44	1.75	−3.53 **	−2.91 **	1.57 **	1.40 **	1.55 **	1.79 **	0.54	0.78 *
LSD Sij _0_._05_	3.93	3.68	0.51	0.73	3.11	2.93	1.89	1.71	2.12	2.00	0.64	0.68	0.43	0.39	0.83	0.75	0.63	0.68
LSD Sij _0_._01_	5.21	4.88	0.68	0.96	4.13	3.89	2.51	2.27	2.81	2.65	0.85	0.90	0.57	0.52	1.10	0.99	0.84	0.91

* and ** indicate *p*-value < 0.05 and 0.01, respectively.

## Data Availability

The data presented in this study are available upon request from the corresponding author.

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
