# Peer review of "Genetic Potential and Inheritance Patterns of Physiological, Agronomic and Quality Traits in Bread Wheat under Normal and Water Deficit Conditions"

_plants, 2022, doi:10.3390/plants11070952_

Round 1

Reviewer 1 Report

The manuscript from Kamara and colleagues evaluated diverse parental bread wheat genotypes and their cross combinations under normal conditions and water stress, in terms of physiological, agronomic and quality traits. The authors presented a large number of data. Nevertheless, some major issues need to be further solved.

One criticism is the presentation of the molecular data related to DREB genes. It is not clear the meaning of these data about one gene. In the present form, these data do not give any additional information about the wheat genotypes. I think that these data should be delated and the manuscript should be addressed specifically to physiological, agronomic and quality traits. The title should be modified.

The data presentation is not suitable. The description of results is too long and not focused. The figure 5, 6, 7 and 8 can be moved as supplementary materials. I suggest to order the genotypes in these figures following the belonging of the group A-E of the cluster dendrogram (fig. 9) to help the reader to follow the different traits as explained in the text (“The drought-tolerant genotypes (groups A and B) proved a substantial increase in physiological traits like chlorophyll a, chlorophyll b, total chlorophyll content, Fv/Fm, relative water content, and proline content by 12.0%, 13.8%, 12.6%, 7.3%, 7.5%, and 14.1%, respectively, compared with the genotypes in group (E). Furthermore, they showed a significant in-crease in the agronomic traits consisting plant height, spike length, number of grains per spike, 1000 grain weight and grain yield by 18.8%, 12.3%, 27.3%, 11.2%, and 51.7% in comparison with the genotypes in group (E).). The description of the results related to figures 5, 6, 7 and 8 is not suitable in this form. I suggest to avoid these long lists of description and focus only on the principal differences that selected genotypes showed. In addition, I am not sure that some of the differences reported in the text are statistically valid (for example, the chlorophyll a content under well-watered condition in Figure 5A is very similar in all the genotypes. Slow differences are visible for P6, P1xP5, P2XP7, P5XP7 and P6XP8 but no statistical analysis is reported).

I suggest to divide Figure 3 in two separate figures for i) Physiological and Biochemical Traits, ii) agronomic and grain quality traits, and insert them after the box plot figures related to the two different class of traits (the actual figures 2 and 4).

The calculated tolerance indices are not reported in the manuscript (“The genotypes in group (A) included two genotypes (P3×P8 and P4×P8) that possessed the highest tolerance indices”).

Figure S1 must to be cancelled. This figure is identical to Figure 10, and the legend is the same to those reported for Figure 11.

In addition, the discussion is not clear in some part. In particular, it is difficult to understand the reference to specific results (for example, the sentence “The parents P3 and P7 were recognized as good combiners for improving proline content under stressed conditions. Moreover, P2, P3, P6 and P8 are promising combiners for most of the studied antioxidant enzymes, chlorophyll content and relative water content” should be make clearer).

Some minor modifications are required, as listed below.

- In the abstract, the sentence “Drought conditions in soil are a serious problem” does not sound well. Please use a better English language. It is well known that water stress positively affects some traits such as proline content or antioxidant enzyme activities (CAT, POD, and SOD). Therefore, the used adverb "nevertheless" in the sentence “proline con-tent, antioxidant enzyme activities (CAT, POD, and SOD), grain protein content, wet gluten content, and dry gluten content were considerably increased compared to well-watered conditions” is not suitable.

- I suggest to add “bread wheat” or “Triticum aestivum” among the keyword

- In materials and Methods, (paragraph 2.1), the number of the genotypes are different from those reported in Table S1: the local genotypes are three, not four, the exotic genotypes form CIMMYT are four, not three.  

- I suggest to report more details about the methods used for hierarchical clustering (paraghraph 2.3)

- The accession numbers of the genes are not present in Pubmed. Anyway, all the parts related to gene sequences have to be deleted.

Author Response

Reviewer 1:

Comments and Suggestions for Authors

The manuscript from Kamara and colleagues evaluated diverse parental bread wheat genotypes and their cross combinations under normal conditions and water stress, in terms of physiological, agronomic and quality traits. The authors presented a large number of data. Nevertheless, some major issues need to be further solved.

One criticism is the presentation of the molecular data related to DREB genes. It is not clear the meaning of these data about one gene. In the present form, these data do not give any additional information about the wheat genotypes. I think that these data should be deleted and the manuscript should be addressed specifically to physiological, agronomic, and quality traits. The title should be modified.

Re: We would like to thank the Reviewer for his time dedicated to our manuscript. The dehydration responsive element-binding (DREB) gene family is a part of the plant-specific family of transcription factors (TFs) which plays a crucial role in regulating and enhancing the expression of numerous stress-responsive genes to abiotic stresses including drought, salinity, and high temperature. From the sequence data, we tried to differentiate between the parental genotypes based on the well-known tolerant genotypes in Genbank.  

After your permission, we would prefer to keep the phylogenetic tree generated based on the sequences that come from DREB gene as a part of biodiversity between evaluated parental genotypes. The alignment sequences (Figure 10) has been moved to supplementary materials (Figure S5). In this respect, many published papers in literature applied DREB and its expression in many plants as an indicator for tolerance to abiotic stresses, as Wang, et al., 2010 (Plant Mol. Biol. Rep. 28: 664-675); Zhao et al., 2014 (Mol. Biol. Rep. 41:1577-1590); Hichri, et al., 2016 (Plant Cell Environ. 39:62-79); Dong et al., 2018 (Front. Plant Sci., 9: 713); Zhou et al., 2020 (Plant Breed. 139: 1158-1167). Here is the link for some of these interested papers:

https://link.springer.com/article/10.1007/s11105-010-0196-y

https://www.readcube.com/articles/10.3389/fpls.2018.00713

https://onlinelibrary.wiley.com/doi/abs/10.1111/pbr.12867

https://link.springer.com/article/10.1007/s11033-013-3004-6

https://onlinelibrary.wiley.com/doi/epdf/10.1111/pce.12591

The data presentation is not suitable, the description of results is too long and not focused. The figure 5, 6, 7 and 8 can be moved as supplementary materials. I suggest to order the genotypes in these figures following the belonging of the group A-E of the cluster dendrogram (fig. 9) to help the reader to follow the different traits as explained in the text.

Re: Figures 5-8 and figure 11 have been moved to the supplementary materials as suggested to be Figure S1-S5. Moreover, the subsection “3.8. Heterosis Relative to Better-Parent” and its corresponding tables have been deleted. The text has been revised and repeated parts have been deleted. In respect of genotype order, the main performance of evaluated genotypes in figures 5-8 are presented before their classifying based on drought tolerance in Figure 9 (Figure 5 in revised version). Accordingly, the genotypes in Figures 5-8 were ordered as commonly used the parents followed by their corresponding crosses.

“The drought-tolerant genotypes (groups A and B) proved a substantial increase in physiological traits like chlorophyll a, chlorophyll b, total chlorophyll content, Fv/Fm, relative water content, and proline content by 12.0%, 13.8%, 12.6%, 7.3%, 7.5%, and 14.1%, respectively, compared with the genotypes in group (E). Furthermore, they showed a significant in-crease in the agronomic traits consisting plant height, spike length, number of grains per spike, 1000 grain weight and grain yield by 18.8%, 12.3%, 27.3%, 11.2%, and 51.7% in comparison with the genotypes in group (E).). The description of the results related to figures 5, 6, 7 and 8 is not suitable in this form. I suggest to avoid these long lists of description and focus only on the principal differences that selected genotypes showed. In addition, I am not sure that some of the differences reported in the text are statistically valid (for example, the chlorophyll a content under well-watered condition in Figure 5A is very similar in all the genotypes. Slow differences are visible for P6, P1xP5, P2XP7, P5XP7 and P6XP8 but no statistical analysis is reported).

Re: This part has been deleted as suggested. The statistical analysis for chlorophyll a content was repeated under well-watered conditions. The result was significant which was prpbably due to the low values of some genotypes as P5×P7 compared to the higher values in the other genotypes.

I suggest to divide Figure 3 in two separate figures for i) Physiological and Biochemical Traits, ii) agronomic and grain quality traits, and insert them after the box plot figures related to the two different class of traits (the actual figures 2 and 4).

Re: The Figure has been divided as suggested to be (A) for physiological and biochemical traits and (B) for agronomic, and grain quality traits. Please see lines 326-327.

The calculated tolerance indices are not reported in the manuscript (“The genotypes in group (A) included two genotypes (P3×P8 and P4×P8) that possessed the highest tolerance indices”).

Re: The tolerance indices have been added to the supplementary materials (Table S3)

Figure S1 must to be cancelled. This figure is identical to Figure 10, and the legend is the same to those reported for Figure 11.

Re: Figure S1 has been deleted as suggested and Figure 10 has been moved to supplementary material with corrected legend (Figure S5).

In addition, the discussion is not clear in some part. In particular, it is difficult to understand the reference to specific results (for example, the sentence “The parents P3 and P7 were recognized as good combiners for improving proline content under stressed conditions. Moreover, P2, P3, P6 and P8 are promising combiners for most of the studied antioxidant enzymes, chlorophyll content and relative water content” should be make clearer.

Re: All discussion section has been revised and mentioned sentences have been modified (lines 572-575).

 Some minor modifications are required, as listed below.

- In the abstract, the sentence “Drought conditions in soil are a serious problem” does not sound well. Please use a better English language. It is well known that water stress positively affects some traits such as proline content or antioxidant enzyme activities (CAT, POD, and SOD). Therefore, the used adverb "nevertheless" in the sentence “proline content, antioxidant enzyme activities (CAT, POD, and SOD), grain protein content, wet gluten content, and dry gluten content were considerably increased compared to well-watered conditions” is not suitable.

Re: The Abstract has been revised and modified as suggested please see lines 26-30 and 37-39

- I suggest to add “bread wheat” or “Triticum aestivum” among the keyword

Re: “Triticum aestivum” has been added to the keyword (line 55)

- In materials and Methods, (paragraph 2.1), the number of the genotypes are different from those reported in Table S1: the local genotypes are three, not four, the exotic genotypes form CIMMYT are four, not three.  

Re: The local genotypes (four genotypes) are Gemmeiza-12, Sids-12, Misr-2 and Gemmeiza-7. Since Misr-2 was imported from CIMMYT as an advanced breeding line but was registered in Egypt as a commercial cultivar.

- I suggest to report more details about the methods used for hierarchical clustering (paraghraph 2.3)

Re: More details have been added, please see lines 209-212

- The accession numbers of the genes are not present in Pubmed. Anyway, all the parts related to gene sequences have to be deleted.

Re: The accession numbers of genes had been removed from the manuscript. These accession numbers will be released to public after publication since I may choosed to release after publication during submission.

Reviewer 2 Report

The manuscript "Genetic Differentiation Based on Dehydration-Responsive Element and Patterns of Physiological, Agronomic and Quality Traits in Bread Wheat in Response to Drought Conditions" compared many traits in eight parental lines and their half-diallel F1s under well watered and water-deficit conditions. Results from this research could be useful for determining the better crosses among eight parental lines. However, the manuscript is very tough to read because too much detailed information makes it too long and tiresome to read, particularly the part of results already had all figures and tables included, but the detailed description from each figure and table actually repeated each content and makes it very hard to draw conclusions from each paragraph. I would suggest the authors to move some unnecessary pictures and tables as the supplemental documents, which will make your results much cleaner and more significant. Description of each table or figure will focus on just the general results, no need to describe every content in such a detail since the information is already supplied in the figures or tables. In one word, the authors need to make a major revision to make the manuscript more concise and the important points more stand out.

Author Response

Reviewer 2:

Comments and Suggestions for Authors

The manuscript "Genetic Differentiation Based on Dehydration-Responsive Element and Patterns of Physiological, Agronomic and Quality Traits in Bread Wheat in Response to Drought Conditions" compared many traits in eight parental lines and their half-diallel F1s under well watered and water-deficit conditions. Results from this research could be useful for determining the better crosses among eight parental lines. However, the manuscript is very tough to read because too much detailed information makes it too long and tiresome to read, particularly the part of results already had all figures and tables included, but the detailed description from each figure and table actually repeated each content and makes it very hard to draw conclusions from each paragraph. I would suggest the authors to move some unnecessary pictures and tables as the supplemental documents, which will make your results much cleaner and more significant. Description of each table or figure will focus on just the general results, no need to describe every content in such a detail since the information is already supplied in the figures or tables. In one word, the authors need to make a major revision to make the manuscript more concise and the important points more stand out.

Re: We would like to thank the Reviewer for his time devoted to our manuscript, and his positive assessment of our work. The manuscript has been meticulously revised, Figures 5-8 and figure 11 have been moved to the supplementary materials as suggested to be Figure S1-S5. Moreover, the subsection “3.8. Heterosis Relative to Better-Parent” and its corresponding tables have been deleted. The text has been revised and repeated parts have been deleted.

Reviewer 3 Report

Following is my review of the article Genetic Differentiation Based on Dehydration-Responsive Element and Patterns of Physiological, Agronomic and Quality Traits in Bread Wheat in Response to Drought Conditions, along my observations are:
The abstract must contain information about the authors' crucial discoveries, as well as information about other significant results.
A precise hypothesis must be presented in the introduction, and the second paragraph of this document must be considerably developed.

Figures 5-8 are difficult to interpret authors should think of other ways to present this data.
Generally speaking, there is a reiteration of material that may have been avoided.
Check the ligands in the figure; they have been written hastily.
A greater amount of material and references should be included in the discussion on relevant and related works.
Restructure and meticulously revise the part on the conclusion based on only significant results.

Author Response

Reviewer 3:

Comments and Suggestions for Authors

Following is my review of the article Genetic Differentiation Based on Dehydration-Responsive Element and Patterns of Physiological, Agronomic and Quality Traits in Bread Wheat in Response to Drought Conditions, along my observations are:
The abstract must contain information about the authors' crucial discoveries, as well as information about other significant results.

Re: We would like to thank the Reviewer for his time dedicated to our manuscript. The abstract has been revised and more information has been added.

A precise hypothesis must be presented in the introduction, and the second paragraph of this document must be considerably developed.

Re: The introduction section has been thoroughly revised and the hypothesis has been added in lines 115-118.

Figures 5-8 are difficult to interpret authors should think of other ways to present this data.
Generally speaking, there is a reiteration of material that may have been avoided.
Check the ligands in the figure; they have been written hastily.
A greater amount of material and references should be included in the discussion on relevant and related works. Restructure and meticulously revise the part on the conclusion based on only significant results.

Re: The results section has been carefully revised, Figures 5-8 and figure 11 have been moved to the supplementary materials as suggested to be Figure S1-S5. Moreover, the subsection “3.8. Heterosis Relative to Better-Parent” and its corresponding tables have been deleted. The text has been revised and repeated parts have been deleted. The ligands have been revised as suggested. The discussion has been revised and modified, more references have been added. The conclusion has been revised and modified as suggested.

Round 2

Reviewer 1 Report

In my previous reviw report, I requested to delete the data related to DREB genes, since these data do not give any additional information about the wheat genotypes and there is no a biological reason to present these data in the present manuscript. I repeat my request to delete these data and to specifically address the manuscript to physiological, agronomic and quality traits. The title have be modified, avoiding the reference to DREB genes.

In addition, the meaning of DREB acronym is "dehydration responsive element binding" not "dehydrin-responsive element-binding", as wrongly reported by the authors.

Author Response

Reviewer 1:

Comments and Suggestions for Authors

In my previous reviw report, I requested to delete the data related to DREB genes, since these data do not give any additional information about the wheat genotypes and there is no a biological reason to present these data in the present manuscript. I repeat my request to delete these data and to specifically address the manuscript to physiological, agronomic and quality traits. The title have be modified, avoiding the reference to DREB genes.

Re: We would like to thank the Reviewer for his time dedicated to our manuscript. We have deleted the data related to the DREB gene and the title has been modified as suggested.

Reviewer 2 Report

The manuscript is still need further revisions:

  1. suggest  to change title to "Evaluation of Physiological, Agronomic and Quality Traits under drought and Analysis of Dehydrin Gene in Eight Bread Wheat Genotypes"
  2. The results part still needs to reduce sentences, in the attached document, I showed the example of removing some unnecessary sentences in section 3.2.1, 3.2.2 and 3.3.3. I'm not requesting authors to change that exactly as I did, but just want to give authors a clue that some sentences can be deleted to make the manuscript cleaner.
  3. The section of 3.5 and 3.6 are both a long paragraph, it can be easily to break them into several paragraph with each short paragraph focus on fewer research points.

Author Response

Reviewer 2:

Comments and Suggestions for Authors

The manuscript still needs further revisions:

  1. suggest to change title to "Evaluation of Physiological, Agronomic and Quality Traits under drought and Analysis of Dehydrin Gene in Eight Bread Wheat Genotypes".

Re: We would like to thank the Reviewer for his time devoted to our manuscript. The title has been modified as suggested with slight modification.

  1. The results part still needs to reduce sentences, in the attached document, I showed the example of removing some unnecessary sentences in section 3.2.1, 3.2.2 and 3.3.3. I'm not requesting authors to change that exactly as I did, but just want to give authors a clue that some sentences can be deleted to make the manuscript cleaner.

Re: The result section has been modified as suggested with minor mention of the promising genotypes under stressed conditions (lines 276-345).

  1. The section of 3.5 and 3.6 are both a long paragraph, it can be easily to break them into several paragraph with each short paragraph focus on fewer research points.

Re: The suggested paragraphs have been divided into subtitles please see 395-465

Reviewer 3 Report

The authors have significantly improved the manuscript; therefore, it can be accepted after a careful English language check.

Author Response

Re: We would like to thank the Reviewer for his time dedicated to our manuscript. For english language check, the manuscript has been revised by our colleague Dr. Ding Zheli, Chinese Academy of Tropical Agricultural Sciences.

Round 3

Reviewer 1 Report

Accepted

Author Response

The authors have significantly improved their manuscript in accordance with the reviewers' suggestions. However, they should further modify the title to more closely follow the recommendation of Reviewer 2. Moreover, they should check the manuscript for typos.

Re: We would like to thank you for the time dedicated to our manuscript. Reviewer 2 suggested to change the title to "Evaluation of Physiological, Agronomic and Quality Traits under drought and Analysis of Dehydrin Gene in Eight Bread Wheat Genotypes". Although, we have deleted the part of the dehydrin-responsive element-binding protein (DREB) from the manuscript based on the request of reviewer 1. Also, this title will cover the agronomic work in the manuscript only, but our work include breeding experiments too. Taking into consideration the suggestion of Reviewer 2 and the parts that have been deleted from the manuscript, the title has been modified. The manuscript has been revised for typos, and it will be further revised for typos by the MDPI team before publishing.

Thanks so much for your kind cooperation

Yours sincerely,

Authors